# IMAGE2SENTENCE BASED ASYMMETRIC ZERO-SHOT COMPOSED IMAGE RETRIEVAL

**Yongchao Du**[1], **Min Wang**[2*], **Wengang Zhou**[1,2*], **Shuping Hui**[1], **Houqiang Li**[1,2]

[1]CAS Key Laboratory of Technology in GIPAS, University of Science and Technology of China
[2]Institute of Artificial Intelligence, Hefei Comprehensive National Science Center
{ycdu2020,huisp}@mail.ustc.edu.cn, wangmin@iai.ustc.edu.cn,
{zhwg,lihq}@ustc.edu.cn

## ABSTRACT

The task of composed image retrieval (CIR) aims to retrieve images based on the query image and the text describing the users' intent. Existing methods have made great progress with the advanced large vision-language (VL) model in CIR task, however, they generally suffer from two main issues: lack of labeled triplets for model training and difficulty of deployment on resource-restricted environments when deploying the large vision-language model. To tackle the above problems, we propose Image2Sentence based Asymmetric zero-shot composed image retrieval (ISA), which takes advantage of the VL model and only relies on unlabeled images for composition learning. In the framework, we propose a new adaptive token learner that maps an image to a sentence in the word embedding space of VL model. The sentence adaptively captures discriminative visual information and is further integrated with the text modifier. An asymmetric structure is devised for flexible deployment, in which the lightweight model is adopted for the query side while the large VL model is deployed on the gallery side. The global contrastive distillation and the local alignment regularization are adopted for the alignment between the light model and the VL model for CIR task. Our experiments demonstrate that the proposed ISA could better cope with the real retrieval scenarios and further improve retrieval accuracy and efficiency.

## 1 INTRODUCTION

Image retrieval is a crucial task for computer vision, and plays a fundamental role in a variety of applications, *e.g.*, product retrieval Dong et al. (2022); Zhan et al. (2021); Liu et al. (2016); Zhou et al. (2010), landmark retrieval Radenović et al. (2018), and person re-identification Zheng et al. (2016); Gheissari et al. (2006). Composed image retrieval (CIR) is presented as a more challenging task that requires accurate understanding of the users' intentions. CIR allows multi-modal queries and aims to retrieve images that are both visually similar to the given query image and match with the text modifier, which enables users to conduct more precise and flexible search. However, to learn the composition ability, it requires labeled triplets of reference image, target image and text modifier Liu et al. (2021); Wu et al. (2021), which are time-consuming and expensive to collect.

To eliminate the limitation of data scarcity, zero-shot composed image retrieval (ZSCIR) is proposed in Saito et al. (2023); Baldrati et al. (2023a), which converts the CIR task to text-to-image retrieval by mapping the image into a single word and combining it with text modifier. However, they all adopt a symmetric retrieval setting based on CLIP models Radford et al. (2021), *i.e.*, using large models to extract both query and database features, and fail to consider the real-life application where the model may be deployed on a resource-limited platform such as mobile device, etc. Moreover, the mapping is implemented by an MLP that projects image feature to a single word vector, *i.e.*, representing image as a single word in the word embedding space. This may be insufficient as a faithful representation for an image, and the heterogeneity of word embedding space and visual feature space may further dampen the discriminative information of image feature after projection, thus constraining its potential.

---

*Corresponding authors: Min Wang and Wengang Zhou

To address the above issues, we propose Image2Sentence based Asymmetric ZSCIR that utilizes different models to extract query and database features. Specifically, a lightweight model is adopted for the query side, which is usually user's terminal device with limited computation resources; the large VL model is employed for the gallery side, which is usually deployed on a cloud platform. Asymmetric structures often suffer from two core issues: inferior representation capability and misalignment between the lightweight model and the larger model. To this end, we propose an adaptive token learner to realize the alignment between the lightweight visual model and the large visual language model, which could further improve the representation discrimination ability of the lightweight model. Specifically, the lightweight model extracts the visual feature map of an input image, then the adaptive token learner extracts visual tokens from visual feature map, and maps them to a more informative sentence in the word embedding space. By adaptively focusing on more distinctive visual patterns, our adaptive token learner serves as a semantic filter that automatically filters out trivial background noise and preserves discriminative visual information. The mapped sentence is learned to cope with the visual feature from the large visual encoder under the global contrastive distillation (GCD), as a constraint to build up semantic alignment between the large model and the lightweight model. Therefore during inference, the mapped sentence is directly concatenated with the modifier text as a caption to search for images. To better learn the image2sentence mapping and retain more visual information, we further leverage the local alignment regularization (LAR) to help the mapped sentence capture more distinctive visual concepts, and to be closer to the real caption that faithfully describes the image. Our asymmetric framework allows for more flexible deployment while also enhancing performance compared with symmetric setting, and outperforms the state-of-the-art methods on three benchmarks.

## 2 RELATED WORK

### 2.1 COMPOSED IMAGE RETRIEVAL

CIR is the task of retrieving images that correspond to the reference image with a modifier text, and has been investigated in several domains, *e.g.*, natural scene Baldrati et al. (2023a); Liu et al. (2021), fashion domain Dodds et al. (2022); Lee et al. (2021); Wu et al. (2021), and synthetic domain Gu et al. (2023); Vo et al. (2019). Supervised learning serves as a stepping stone in the progression of CIR, and is first introduced in Vo et al. (2019), in which meticulously annotated (reference image, text modifier, target image) triplets are adopted for training. Supervised methods mainly follow the pipeline that merges the features of reference image and text modifier to align with the target image feature, as a result, their innovations generally revolve around jointly capturing the information from both visual and textual modality Vo et al. (2019); Lee et al. (2021); Chen et al. (2020). Moreover, some methods further improve the performance by adopting multimodal foundation model Baldrati et al. (2021; 2022a;b; 2023b); Levy et al. (2023) or incorporating pretraining-finetuning strategy Dodds et al. (2022); Han et al. (2022; 2023).

However, the above supervised methods heavily rely on the triplet training dataset, which involves great difficulty in collecting and labeling in large scale. A new task of zero-shot composed image retrieval (ZSCIR) Saito et al. (2023) is proposed to liberate the model learning from meticulous triplet-labeled data, and relies only on image-text pairs. Saito et al. (2023) propose to use MLP to convert CLIP visual feature to a single word in the word embedding space, and adopt a cycle contrastive loss for training; Baldrati et al. (2023a) propose a two-stage framework including text inversion and distillation to convert image into text domain. However, the above methods rely on symmetric retrieval, in which heavy models are utilized to extract features for both query and database, and are difficult to deploy in practical retrieval application scenarios. We propose a ZS-CIR framework, which enables flexible deployment on resource-limited scenarios by adopting the lightweight model. Different from previous works, instead of converting image to a single word, we propose the adaptive token learner to filter out more discriminative visual information, which is further mapped to a more informative sentence. Apart from global contrastive loss, a dense local alignment regularization is applied on mapped sentences to preserve the visual information during conversion. Our work shows its superiority in both retrieval accuracy and deployment flexibility.

### 2.2 ASYMMETRIC RETRIEVAL

In recent times, the adoption of expansive deep models has greatly improved the multi-modal comprehension ability Jia et al. (2021); Li et al. (2021; 2022; 2023); Radford et al. (2021). However,

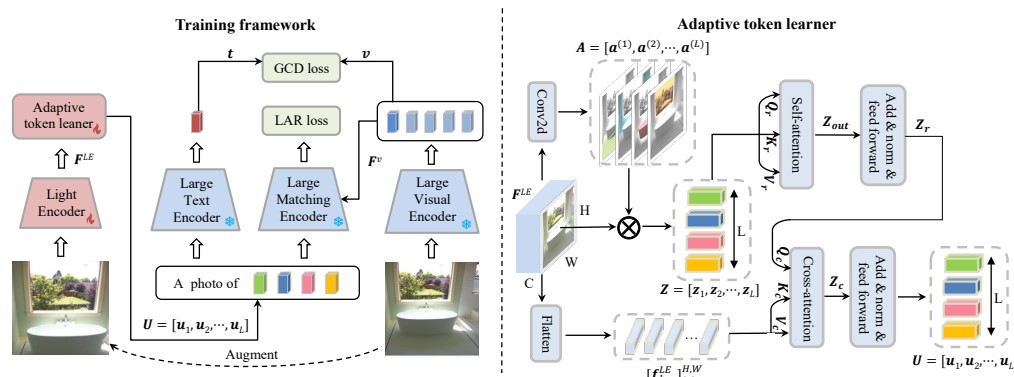

Figure 1: **Left**: our training framework. Large visual encoder, large text encoder and matching encoder of pre-trained foundation model are frozen during training. The framework is trained with global contrastive distillation (GCD) and local alignment regularization (LAR). **Right**: adaptive token learner. Adaptive token learner learns to adaptively select salient visual patterns and map them to a sentence in the word embedding space of text encoder with attention mechanism.

large pre-trained models usually consume more resources in computing and memory, which is not desirable for some resource-constrained scenarios such as image retrieval. To tackle this inconvenience, recently many works Budnik & Avrithis (2021); Wu et al. (2022a); Duggal et al. (2021); Wu et al. (2022b) have been proposed to research into asymmetric image retrieval, in which the query (user) side adopts a lightweight model such as Howard et al. (2017; 2019); Liu et al. (2023); Mehta & Rastegari (2021); Sandler et al. (2018); Tan & Le (2019) while the gallery (server) side takes a large model. To our knowledge, our proposed framework is the first work to adopt asymmetric retrieval in CIR task, and could even outperform its symmetric counterpart.

## 3 METHOD

The overview of our framework is shown in Figure 1. Our framework integrates a pre-trained vision-language (VL) model, comprising a large visual encoder and a large text encoder. These encoders are specifically designed to align visual and textual modalities within a unified feature space. Additionally, a matching encoder is incorporated to enhance image-text matching capabilities, which ensures fine-grained alignment and interpretation between different modalities. We freeze the parameter of VL model to maintain the alignment between visual and textual modality during training. Given an image $I$, a lightweight visual encoder is applied to extract visual feature map $F^{LE}$ of augmented $I$, and a new adaptive token learner is utilized to map the visual feature map to a series of sentence tokens $U$. $U$ is then concatenated with a prompt such as "a photo of" to form a full sentence, which is fed into the large text encoder to obtain textual feature $t$. Large visual encoder is utilized to extract the image feature $v$ and feature map $F^v$ of $I$. Global contrastive distillation is applied to align $v$ and $t$ while local alignment regularization is applied on $U$ and $F^v$.

### 3.1 LIGHT VISUAL ENCODER AND ADAPTIVE TOKEN LEARNER

As shown in Figure 1, we apply data augmentation to input image $I$ and feed it into the light visual encoder, to extract visual feature map $F^{LE} \in \mathbb{R}^{H \times W \times C}$. It is a common understanding that light models are less discriminative compared with the large models. Therefore, we introduce an innovative adaptive token learner. Instead of representing an image as a single word, our method conceptualizes the image as a sentence, which significantly enhances visual discrimination within the word embedding space. Each image element is treated as an individual narrative component, allowing for a richer and more detailed representation in the embedding space. Specifically, we map each pixel $f_{h,w}^{LE}$ of $F^{LE}$ to one of $L$ semantic groups using convolutions, and generate $L$ attention maps $A = \{a^{(1)}, \cdots, a^{(L)}\}$ conditioned on the $F^{LE}$. The spatial attention maps are computed as

$$a_{h,w}^{(l)} = \frac{\exp(w^{(l)} \cdot f_{h,w}^{LE})}{\sum_{i=1}^{L} \exp(w^{(l)} \cdot f_{h,w}^{LE})},$$ (1)

where $W = [w^{(1)}, \cdots, w^{(L)}] \in \mathbb{R}^{L \times C}$ are the parameters of the $L$ convolutions. Spatial attention is applied to adaptively discover discriminative visual patterns, and obtain $L$ visual tokens $Z = [z^{(1)}, \cdots, z^{(L)}]$

$$\boldsymbol{z}^{(l)} = \frac{1}{\rho(\boldsymbol{a}^{(l)})} \sum_{h=1,w=1}^{H,W} \boldsymbol{a}_{h,w}^{(l)} \cdot \boldsymbol{f}_{h,w}^{LE}, \quad \rho(\boldsymbol{a}^{(l)}) = \sum_{h=1,w=1}^{H,W} \boldsymbol{a}_{h,w}^{(l)}. \tag{2}$$

Since each visual token is learned separately without considering their mutual relationship, self-attention is utilized to model the interaction among visual tokens to obtain the refined tokens $\boldsymbol{Z}_r$

$$\begin{aligned} \boldsymbol{Z}_{out} &= \boldsymbol{Z} + \mathrm{softmax}_L((\boldsymbol{Z}\boldsymbol{K}_r)(\boldsymbol{Z}\boldsymbol{Q}_r)^T)\boldsymbol{Z}\boldsymbol{V}_r, \\ \boldsymbol{Z}_r &= \boldsymbol{Z}_{out} + \sigma(\boldsymbol{Z}_{out}\boldsymbol{W}_{r1})\boldsymbol{W}_{r2}, \end{aligned} \tag{3}$$

where $\boldsymbol{Q}_r, \boldsymbol{K}_r, \boldsymbol{V}_r$ are the parameters of query, key and value, and $\boldsymbol{W}_{r1}, \boldsymbol{W}_{r2}$ are the parameters of feed-forward network with $\sigma(\cdot)$ activation. To obtain the $L$ sentence tokens that involve filtered visual semantics, we adopt the cross-attention to adaptively select the salient visual patterns from the feature map $\boldsymbol{F}^{LE}$. $\boldsymbol{F}^{LE}$ is first flattened as a sequence $\boldsymbol{F}^{LE'} = [\boldsymbol{f}_{1,1}^{LE}, \cdots, \boldsymbol{f}_{H,W}^{LE}] \in \mathbb{R}^{HW \times C}$, and the sentence tokens are calculated as

$$\begin{aligned} \boldsymbol{Z}_c &= \boldsymbol{Z}_r + \mathrm{softmax}_L((\boldsymbol{F}^{LE'}\boldsymbol{K}_c)(\boldsymbol{Z}_r\boldsymbol{Q}_c)^T)\boldsymbol{F}^{LE'}\boldsymbol{V}_c, \\ \boldsymbol{U} &= [\boldsymbol{Z}_c + \sigma(\boldsymbol{Z}_c\boldsymbol{W}_{c1})\boldsymbol{W}_{c2}]\boldsymbol{W}_p, \end{aligned} \tag{4}$$

where $\boldsymbol{Q}_c, \boldsymbol{K}_c, \boldsymbol{V}_c$ are the parameters of query, key and value, and $\boldsymbol{W}_{c1}, \boldsymbol{W}_{c2}$ are the parameters of feed-forward network with $\sigma(\cdot)$ activation. $\boldsymbol{W}_p \in \mathbb{R}^{C \times d_w}$ are the parameters of fc layer to project the output from feed-forward network to $L$ sentence tokens $\boldsymbol{U} = [\boldsymbol{u}^{(1)}, \cdots, \boldsymbol{u}^{(L)}] \in \mathbb{R}^{L \times d_w}$, where $d_w$ is the dimension of word embedding space. In sum, the adaptive token learner incorporates the spatial attention mechanism to filter out noisy information by selecting more prominent visual patterns, and map them to the word embedding space as a sentence.

## 3.2 LEARNING WITH PRE-TRAINED FOUNDATION MODEL

Given sentence tokens $\boldsymbol{U}$, we concatenate them with the predefined prompt to form a full sentence, and feed it into the large text encoder to obtain the textual feature $\boldsymbol{t}$. During training, $\boldsymbol{t}$ is constrained to be similar to the corresponding image feature $\boldsymbol{v}$ extracted from the large visual encoder within batch $\mathcal{B}$. The normalized image-to-text and text-to-image similarity are calculated as

$$\boldsymbol{p}^{I2T} = \frac{\exp(\boldsymbol{v}_i^T\boldsymbol{t}_i/\tau)}{\sum\limits_{j \in \mathcal{B}} \exp(\boldsymbol{v}_i^T\boldsymbol{t}_j/\tau)}, \quad \boldsymbol{p}^{T2I} = \frac{\exp(\boldsymbol{t}_i^T\boldsymbol{v}_i/\tau)}{\sum\limits_{j \in \mathcal{B}} \exp(\boldsymbol{t}_i^T\boldsymbol{v}_j/\tau)}, \tag{5}$$

where $\tau$ denotes the temperature parameter. Suppose $\boldsymbol{y}^{I2T}, \boldsymbol{y}^{T2I}$ denote the ground-truth one-hot similarity, the global contrastive distillation is computed as the cross-entropy $H$ between $\boldsymbol{p}$ and $\boldsymbol{y}$

$$\mathcal{L}_{GCD} = \frac{1}{2}\mathbb{E}_{\mathcal{B}}[H(\boldsymbol{p}^{I2T}, \boldsymbol{y}^{I2T}) + H(\boldsymbol{p}^{T2I}, \boldsymbol{y}^{T2I})]. \tag{6}$$

To further enhance the semantic alignment of sentence tokens with visual information, local alignment regularization is applied to harness sentence tokens to be more like the caption that faithfully describes the image content in the word embedding space. Suppose that the word embedding space is shared across the text encoder and matching encoder, therefore the local alignment regularization will help to generate better sentence tokens, and improve the global semantic alignment between $\boldsymbol{t}$ and $\boldsymbol{v}$. Specifically, we feed sentence tokens together with visual feature map $\boldsymbol{F}^v$ into the matching encoder to obtain the probability $\boldsymbol{p}^{LAR}$. The local alignment regularization is calculated as

$$\mathcal{L}_{LAR} = \mathbb{E}_{\mathcal{B}}[H(\boldsymbol{p}^{LAR}, \boldsymbol{y}^{LAR})], \tag{7}$$

where $\boldsymbol{y}^{LAR}$ denotes the binary one-hot vector that represents the paired label. Therefore, the final loss could be computed as

$$\mathcal{L} = \mathcal{L}_{GCD} + \mathcal{L}_{LAR}. \tag{8}$$

With the knowledge from the pre-trained foundation model, our framework could acquire composition ability by training on unlabeled images.

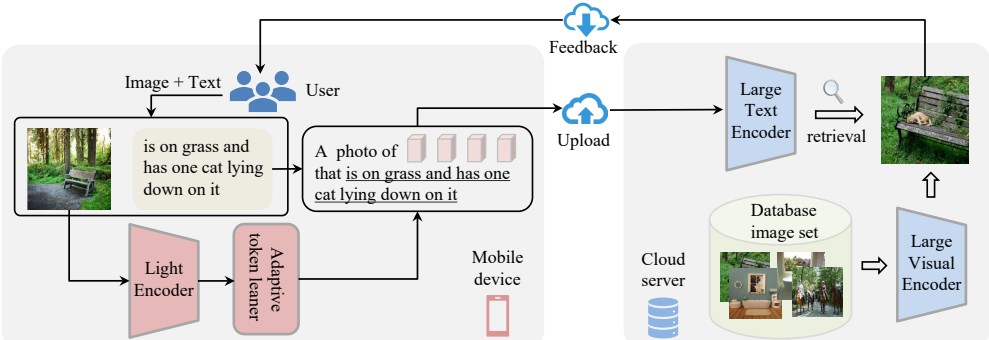

Figure 2: The inference workflow of our framework. The light encoder and the adaptive token learner are deployed on mobile device, to extract sentence tokens from the query image and concatenate them with the prompt and text modifier as the composed query. Pre-trained foundation model is deployed on the cloud server, and the large text encoder receives the uploaded composed query to extract the text feature for retrieval, and the large visual encoder is used to extract features of database images. The retrieval results are returned to the user side.

## 3.3 INFERENCE

In practical application scenarios, the memory and computing resources should be included in our consideration. Therefore, as shown in Figure 2, the lightweight visual encoder and the adaptive token learner are deployed on mobile devices, while the large visual and text encoder are deployed on the server side. During the inference phase, the user inputs a query image along with a text modifier. The system then converts this query image into sentence tokens, a process that encapsulates the visual content into a textual format. This conversion is pivotal for the subsequent retrieval process, as it enables the system to interpret and process the visual data in a linguistically contextualized manner. Specifically, we concatenate the sentence tokens and pre-defined prompts together with text modifier as the composed query: [prompt] $u_1, u_2, \cdots, u_L$ that [text modifier]. It is uploaded to the cloud server to retrieve the images from the database. The large text encoder is utilized to extract the composed query feature during online inference, and the large visual encoder is only utilized to extract image features from the database offline.

## 4 EXPERIMENTS

In this section, we conduct a series of experiments to evaluate our ISA on the task of zero-shot composed image retrieval. Three datasets are utilized to verify the effectiveness of our method, including CIRR Liu et al. (2021), FashionIQ Wu et al. (2021) and CIRCO Baldrati et al. (2023a).

### 4.1 IMPLEMENTATION DETAIL

We adopt the BLIP model Li et al. (2022) pre-trained on 129M image-text pairs as the multi-modal foundation model. CC3M Sharma et al. (2018) is adopted as the training set in our framework.

Since BLIP visual encoder is frozen during training, image feature $v$ and visual feature map $F^v$ could be extracted and stored in advance. For the adaptive token learner, the token length $L$ is set as 6, and two hidden dimensions of feed-forward are set as 256 and 512, respectively. AdamW optimizer with 3e-4 learning rate is adopted, and the framework is trained for 20 epochs with 5 epochs of linear warm-up and 15 epochs of cosine annealing.

| Query model | Gallery model | FLOPS(G) | | PARAM(M) | |
|---|---|---|---|---|---|
| | | ABS | % | ABS | % |
| BLIP VE | BLIP VE | 16.86 | 100.0 | 87.0 | 100.0 |
| EfficientNet B0 | | 0.43 | 2.6 | 5.3 | 6.1 |
| EfficientNet B2 | | 0.72 | 4.3 | 8.5 | 9.7 |
| EfficientViT M2 | BLIP VE | 0.21 | 1.2 | 5.0 | 5.7 |
| MobileNet V2 | | 0.35 | 2.1 | 3.5 | 4.0 |
| MobileViT V2 | | 1.42 | 8.4 | 5.5 | 6.3 |

Table 1: FLOPS and parameters of different light models with adaptive token learner in our work, absolute (ABS) and relative (%) to the BLIP visual encoder (BLIP VE).

The proposed method is implemented on open source Pytorch framework on a server with 4 NVIDIA GeForce RTX 3090 GPU, with the batch size as 320.

We first compare with three simple baselines which directly extract the visual and text features using large foundation model: (i) Image-only, using only visual features for both query and database; (ii) Text-only, using textual features for query and visual features for database; (iii) Image + Text, simply merging the visual and textual features with addition for query, while using the visual features for database. For the state-of-the-art ZSCIR methods, Pic2word Saito et al. (2023)and SEARLE Baldrati et al. (2023a) are included. Since Pic2word and SEARLE are all based on ViT-L/14 CLIP model, we re-implement their methods on BLIP with the open-source codes for fair comparison. Two lightweight models are mainly used as the lightweight visual encoder for our ISA: EfficientNet B2 Tan & Le (2019) and EfficientViT M2 Liu et al. (2023), as representatives of two main categories of deep architectures (CNN and transformer). Since CLIP models do not include the pre-trained matching encoder that share the word embedding space with the text encoder, we only report our results based on BLIP. We also compare ISA with a symmetric baseline, *i.e.*, frozen BLIP visual encoder is adopted to obtain the visual feature map, followed by the same adaptive token learner to obtain the sentence tokens.

| Gallery model | Query model | Method | Recall@K | | | | Avg |
| | | | R@1 | R@5 | R@10 | R@50 | |
| --- | --- | --- | --- | --- | --- | --- | --- |
| CLIP ViT-L/14 | CLIP VE | Image-only | 7.40 | 23.60 | 34.00 | 57.40 | 30.60 |
| | - | Text-only | 20.90 | 44.80 | 56.70 | 79.10 | 50.36 |
| | CLIP VE | Image + Text | 12.40 | 36.20 | 49.10 | 78.20 | 43.97 |
| | CLIP VE | Pic2Word Saito et al. (2023) | 23.90 | 51.70 | 65.30 | 87.80 | 57.17 |
| | CLIP VE | SEARLE Baldrati et al. (2023a) | 24.22 | 52.41 | 66.29 | 88.63 | 57.88 |
| BLIP | BLIP VE | Image-only | 7.52 | 21.83 | 31.21 | 52.94 | 28.37 |
| | - | Text-only | 26.48 | 50.46 | 61.22 | 82.26 | 55.10 |
| | BLIP VE | Image + Text | 8.19 | 24.19 | 34.41 | 57.81 | 31.15 |
| | BLIP VE | Pic2word Saito et al. (2023) | 26.70 | 53.16 | 64.10 | 84.36 | 57.08 |
| | BLIP VE | SEARLE Baldrati et al. (2023a) | 29.27 | 54.86 | 66.57 | 86.16 | 59.21 |
| | BLIP VE | **Ours(Sym)** | 29.68 | 58.72 | 70.79 | 90.33 | 62.38 |
| | EfficientNet B2 | **Ours(Asy)** | **30.84** | **61.06** | **73.57** | **92.43** | **64.48** |
| | EfficientViT M2 | **Ours(Asy)** | 29.63 | 58.99 | 71.37 | 91.47 | 62.87 |

Table 2: Results of zero-shot composed image retrieval on CIRR test set. The best result for each column is shown in bold fonts. Sym: use BLIP visual models to both extract features for query and database images. Asy: use different visual models to extract features for query and database images.

| Gallery model | Query model | Method | mAP@K | | | | Avg |
| | | | mAP@5 | mAP@10 | mAP@25 | mAP@50 | |
| --- | --- | --- | --- | --- | --- | --- | --- |
| CLIP ViT-L/14 | CLIP VE | Image-only | 1.69 | 2.18 | 2.78 | 3.22 | 2.47 |
| | - | Text-only | 2.67 | 3.00 | 3.38 | 3.69 | 3.19 |
| | CLIP VE | Image + Text | 3.60 | 4.40 | 5.59 | 6.21 | 4.95 |
| | CLIP VE | Pic2Word Saito et al. (2023) | 8.43 | 9.11 | 10.15 | 10.73 | 9.61 |
| | CLIP VE | SEARLE Baldrati et al. (2023a) | 10.18 | 11.03 | 12.72 | 13.67 | 11.90 |
| BLIP | BLIP VE | Image-only | 1.28 | 1.48 | 1.88 | 2.13 | 1.69 |
| | - | Text-only | 4.01 | 4.37 | 4.79 | 5.13 | 4.58 |
| | BLIP VE | Image + Text | 1.39 | 1.72 | 2.22 | 2.51 | 1.96 |
| | BLIP VE | Pic2word Saito et al. (2023) | 8.69 | 9.36 | 10.40 | 10.99 | 9.86 |
| | BLIP VE | SEARLE Baldrati et al. (2023a) | 10.65 | 11.34 | 12.40 | 13.02 | 11.85 |
| | BLIP VE | **Ours(Sym)** | 9.67 | 10.32 | 11.26 | 11.61 | 10.72 |
| | EfficientNet B2 | **Ours(Asy)** | **11.33** | **12.25** | **13.42** | **13.97** | **12.74** |
| | EfficientViT M2 | **Ours(Asy)** | 9.82 | 10.50 | 11.61 | 12.09 | 11.01 |

Table 3: Results of zero-shot composed image retrieval on CIRCO test set. The best result for each column is shown in bold fonts. Sym: use BLIP visual models to both extract features for query and database images. Asy: use different visual models to extract features for query and database images.

## 4.2 COMPARISON WITH THE STATE-OF-THE-ART METHODS

We compare different methods on the test set of CIRR and CIRCO while on the validation set of FashionIQ, and the final results are computed as the average of 3 runs on different random seeds for training. As shown in Table 1, the lightweight models are more efficient in parameters and computational complexity compared with BLIP visual encoder. The retrieval performance on three benchmark datasets is reported in Table 2, 3, 4, respectively. Our method could outperform simple baselines (Image-only, Text-only and Image + Text) by a large margin. From the results of Pic2word and SEARLE, we find that the methods based on BLIP are generally better than those based on CLIP for ZSCIR. Our EfficientNet B2-based ISA could outperform Pic2word and SEARLE on three benchmark datasets, while EfficientViT M2-based ISA achieves better results on FashionIQ and CIRR and is comparable with SEARLE on CIRCO. Compared with our symmetric retrieval in

| Gallery model | Query model | Method | Dress | | Shirt | | Top&tee | | Average | |
|---|---|---|---|---|---|---|---|---|---|---|
| | | | R@10 | R@50 | R@10 | R@50 | R@10 | R@50 | R@10 | R@50 |
| CLIP ViT-L/14 | CLIP VE | Image-only | 5.40 | 13.90 | 9.90 | 20.80 | 8.30 | 17.70 | 7.90 | 17.50 |
| | - | Text-only | 13.60 | 29.70 | 18.90 | 31.80 | 19.30 | 37.00 | 17.30 | 32.90 |
| | CLIP VE | Image + Text | 16.30 | 33.60 | 21.00 | 34.50 | 22.20 | 39.00 | 19.80 | 35.70 |
| | CLIP VE | Pic2Word Saito et al. (2023) | 20.00 | 40.20 | 26.20 | 43.60 | 27.90 | 47.40 | 24.70 | 43.70 |
| | CLIP VE | SEARLE Baldrati et al. (2023a) | 20.32 | 43.18 | 27.43 | 45.68 | 29.32 | 50.17 | 25.69 | 46.34 |
| BLIP | BLIP VE | Image-only | 4.07 | 11.35 | 7.07 | 16.39 | 6.88 | 13.67 | 6.01 | 13.67 |
| | - | Text-only | 18.10 | 35.05 | 23.45 | 39.21 | 26.06 | 44.93 | 22.54 | 39.73 |
| | BLIP VE | Image + Text | 5.16 | 13.58 | 8.15 | 18.89 | 7.80 | 16.62 | 7.03 | 16.36 |
| | BLIP VE | Pic2word Saito et al. (2023) | 21.35 | 42.68 | 27.51 | 46.01 | 29.12 | 49.33 | 25.99 | 46.00 |
| | BLIP VE | SEARLE Baldrati et al. (2023a) | 22.11 | 41.79 | 29.72 | 48.53 | 31.03 | 52.37 | 27.62 | 47.56 |
| | BLIP VE | **Ours(Sym)** | 24.69 | 43.88 | **30.79** | **50.05** | **33.91** | 53.65 | **29.79** | 49.19 |
| | EfficientNet B2 | **Ours(Asy)** | 25.33 | **46.26** | 30.03 | 48.58 | 33.45 | 53.80 | 29.60 | **49.54** |
| | EfficientViT M2 | **Ours(Asy)** | **25.48** | 45.51 | 29.64 | 48.68 | 32.94 | **54.31** | 29.35 | 49.50 |

Table 4: Results of zero-shot composed image retrieval on FashionIQ validation set. The best result for each column is shown in bold fonts. Sym: use BLIP visual model to both extract features for query and database. Asy: use different visual models to extract features for query and database.

Ours (Sym), our ISA could achieve superior performance, which could be attributed to ISA containing more training parameters compared with Ours (Sym) (lightweight model + adaptive token learner v.s. adaptive token learner) during training. Meanwhile, ISA consumes much less computation resources during online retrieval and is more flexible to deploy for real application scenarios, demonstrating the effectiveness of our framework.

## 4.3 ABLATION STUDY

For ablation study, we mainly adopt the validation set of CIRR and CIRCO for evaluation, and the results for FashionIQ are included in the appendix.

### 4.3.1 TRAINING WITH DIFFERENT MAPPING NETWORK AND NUMBER OF TOKENS

To verify the effectiveness of our adaptive token learner, we compare it with a variant that adopts MLP as the mapping network, which maps the image feature to a single word following Baldrati et al. (2023a); Saito et al. (2023). The results are shown in Table 5, which demonstrate the consistent superiority for both EfficientNet B2 and EfficientViT M2. These results prove the effectiveness of our adaptive token learner, which maps the visual feature map to multiple sentence tokens per image, and helps to adaptively select more discriminative visual patterns from the query image to generate more informative sentence tokens. It is notable that the variant MLP (single token) could be seen as Pic2word with LAR and outperform Pic2word, thus proving the effectiveness of LAR.

| Query model | Mapping network | CIRR | | | | | CIRCO | | | | |
|---|---|---|---|---|---|---|---|---|---|---|---|
| | | R@1 | R@5 | R@10 | R@50 | Avg | mAP@5 | mAP@10 | mAP@25 | mAP@50 | Avg |
| EfficientNet B2 | ATL | **31.48** | **63.42** | **77.27** | **93.16** | **66.33** | **13.19** | **13.83** | **15.20** | **15.85** | **14.52** |
| EfficientNet B2 | MLP (single token) | 30.18 | 59.46 | 71.30 | 89.33 | 62.57 | 8.28 | 8.67 | 9.62 | 10.00 | 9.14 |
| EfficientViT M2 | ATL | 31.26 | 62.31 | 75.70 | 92.38 | 65.41 | 12.24 | 12.75 | 13.80 | 14.27 | 13.27 |
| EfficientViT M2 | MLP (single token) | 30.32 | 60.25 | 71.49 | 89.57 | 62.90 | 7.19 | 7.73 | 8.47 | 9.09 | 8.12 |

Table 5: Results of different mapping networks with BLIP on CIRR and CIRCO validation set. ATL denotes the proposed adaptive token learner. The best results are shown in bold fonts.

For our adaptive token learner, we show the performance of our method with different token lengths $L$ on three datasets in Figure 3. We average the recall or precision values for each dataset to obtain the overall evaluation criterion. We find that the average performance improves with the increase of token length for both lightweight visual encoders, however excessive large token length leads to performance degradation. We visualize the attention map of adaptive token learner and the retrieval results in Figure 4, and discover that with excessive large token lengths, some sentence tokens may tend to focus on the background or the trivial patterns, which would likely introduce noise to degrade the retrieval performance.

### 4.3.2 TRAIN WITH DIFFERENT QUERY MODELS

In order to investigate the influence of different lightweight encoders, we conduct experiments with different light visual models and pre-training parameters, as shown in Table 6. We could see that EfficientNet B2 shows the best overall performance in CNN, while EfficientViT M2 is proved to be the best in transformer. This could be attributed to the superior network design and larger capacity of

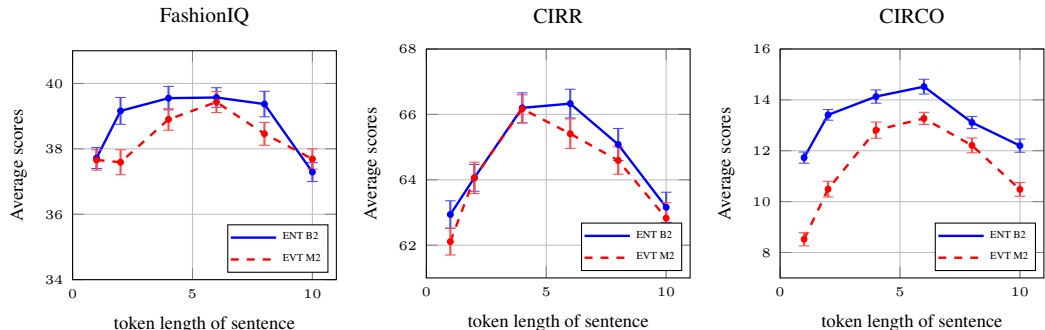

Figure 3: The average and standard deviation of performance scores on three validation sets.

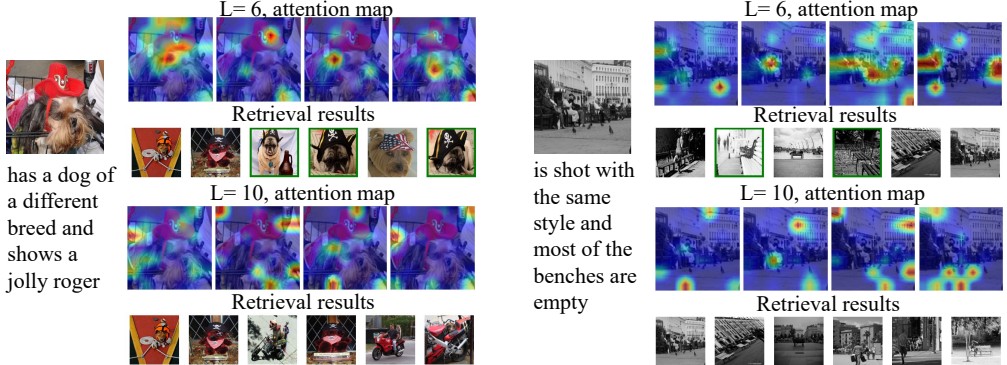

Figure 4: Attention map of sentence tokens for different token lengths and retrieval results. Query image and text modifier are on the left side. Ground-truth target images are marked in green boxes.

parameters. Moreover, better initialization weights, *i.e.*, adopting weights pre-trained on ImageNet Deng et al. (2009), could obtain better performance compared with random initialization. This reveals the possibility of adopting different lightweight models to flexibly meet various efficiency and retrieval accuracy requirements, and improve the performance with better pre-trained weights.

| Query model | CIRR | | | | Avg | CIRCO | | | | Avg |
|---|---|---|---|---|---|---|---|---|---|---|
| | R@1 | R@5 | R@10 | R@50 | | mAP@5 | mAP@10 | mAP@25 | mAP@50 | |
| EfficientNet B0 | **32.02** | **64.57** | 76.12 | 92.29 | 66.25 | 11.11 | 12.04 | 13.14 | 13.67 | 12.49 |
| EfficientNet B0* | 27.82 | 58.38 | 71.16 | 90.07 | 61.86 | 9.5 | 9.96 | 10.6 | 11.02 | 10.27 |
| EfficientNet B2 | 31.48 | 63.42 | **77.27** | **93.16** | **66.33** | **13.19** | **13.83** | **15.20** | **15.85** | **14.52** |
| EfficientNet B2* | 29.47 | 60.13 | 74.05 | 91.27 | 63.73 | 11.16 | 11.57 | 12.56 | 13.13 | 12.11 |
| EfficientViT M2 | 31.26 | 62.31 | 75.70 | 92.38 | 65.41 | 12.24 | 12.75 | 13.80 | 14.27 | 13.27 |
| EfficientViT M2* | 28.56 | 59.46 | 71.11 | 89.38 | 62.13 | 10.19 | 10.79 | 11.48 | 11.94 | 11.10 |
| MobileNet V2 | 30.50 | 63.51 | 74.27 | 91.18 | 64.87 | 10.50 | 10.82 | 12.09 | 12.44 | 11.46 |
| MobileNet V2* | 29.47 | 60.14 | 72.10 | 90.07 | 62.95 | 8.55 | 9.16 | 10.04 | 10.39 | 9.54 |
| MobileViT V2 | 27.63 | 58.66 | 70.85 | 88.68 | 61.46 | 9.44 | 9.68 | 10.49 | 10.96 | 10.14 |
| MobileViT V2* | 27.02 | 57.92 | 69.33 | 88.96 | 60.81 | 8.41 | 8.77 | 9.54 | 9.97 | 9.17 |

Table 6: Results of different query models with BLIP on CIRR/CIRCO validation set. * denotes adopting random initialization for light query model. The best results are shown in bold fonts.

### 4.3.3 SYMMETRIC AND ASYMMETRIC RETRIEVAL

To verify the effectiveness of asymmetric framework, we conduct additional experiments on symmetric setting, which adopts the same model to extract both query and database features. Specifically, for the symmetric retrieval in the second, 4-th and 6-th rows of Table 7, query and gallery models are both updated during training. Therefore, the network should both learn to generate the image feature, and map the image to the sentence tokens in the word embedding space. We find that they would drastically degrade the performance. This could be attributed to the confusion of multi-task learning for image feature and sentence tokens, since image feature space and word embedding space are heterogeneous semantic spaces. Moreover, we also carry out additional experiments that adopting BLIP VE as both query model and gallery model, and only query model is updated during

training to avoid the multi-task confusion. Therefore, BLIP VE is only trained to generate sentence tokens. As revealed in the result of BLIP VE †, there is no breaking down for learning, however it is still outperformed by EfficientNet B2-based ISA. This could be explained by the fact that BLIP VE is much heavier than light models as shown in Table 1, and could be trained only with reduced batch size, thus degrading the effectiveness of contrastive and LAR loss. The above results reveal the importance of pre-trained alignment for different modalities in ZSCIR, but the BLIP visual encoder could be replaced with lightweight model for efficiency.

| Query model | Gallery model | CIRR | | | | | CIRCO | | | | |
|---|---|---|---|---|---|---|---|---|---|---|---|
| | | R@1 | R@5 | R@10 | R@50 | Avg | mAP@5 | mAP@10 | mAP@25 | mAP@50 | Avg |
| EfficientNet B2 | BLIP VE | **31.48** | **63.42** | **77.27** | **93.16** | **66.33** | **13.19** | **13.83** | **15.20** | **15.85** | **14.52** |
| EfficientNet B2 | EfficientNet B2 | 0.61 | 0.74 | 1.02 | 3.97 | 1.59 | 0.00 | 0.02 | 0.08 | 0.14 | 0.06 |
| EfficientViT M2 | BLIP VE | 31.26 | 62.31 | 75.70 | 92.38 | 65.41 | 12.24 | 12.75 | 13.80 | 14.27 | 13.27 |
| EfficientViT M2 | EfficientViT M2 | 0.29 | 0.37 | 0.79 | 2.68 | 1.03 | 0.00 | 0.01 | 0.04 | 0.11 | 0.04 |
| BLIP VE † | BLIP VE | 30.93 | 62.09 | 74.67 | 92.90 | 65.15 | 12.60 | 13.20 | 14.56 | 15.07 | 13.86 |
| BLIP VE | BLIP VE | 0.49 | 0.60 | 0.83 | 2.96 | 1.22 | 0.02 | 0.03 | 0.08 | 0.15 | 0.07 |

Table 7: Comparison with symmetric retrieval on the validation set of CIRR and CIRCO. BLIP VE denotes the BLIP visual encoder. † denotes updating the query BLIP VE while preserving the pre-trained BLIP VE in gallery side during training for maintaining the multi-modality alignment. The best results are shown in bold fonts.

### 4.3.4 LEARNING WITH DIFFERENT LOSS CONSTRAINTS

In our framework, we mainly apply global contrastive distillation (GCD) and local alignment regularization (LAR) for learning. We verify the losses by dropping each of them separately. To further validate LAR, we compare it with the text regularization, *i.e.*, LLM TR proposed in Baldrati et al. (2023a). Specifically, LLM TR is realized following the pipeline that utilizes a classification model to label each training image and an LLM to do sentence-making. A cosine loss is applied to constrain the sentence tokens to be similar to the LLM-made sentence as text regularization. As reported in Table 8, it is obvious that both losses are crucial for training, since dropping either of them would lead to consistent worse performance. Text regularization from LLM-made sentences would be helpful but only restricted to CIRCO, however it needs extra classification model and LLM model which leads to a much more complex pipeline. Our LAR is simple but more effective on three benchmarks.

| Query model | Variants | CIRR | | | | | CIRCO | | | | |
|---|---|---|---|---|---|---|---|---|---|---|---|
| | | R@1 | R@5 | R@10 | R@50 | Avg | mAP@5 | mAP@10 | mAP@25 | mAP@50 | Avg |
| EfficientNet B2 | GCD + LAR | **31.48** | **63.42** | **77.27** | **93.16** | **66.33** | **13.19** | **13.83** | **15.20** | **15.85** | **14.52** |
| | GCD | 29.86 | 60.18 | 73.90 | 91.13 | 63.77 | 10.46 | 10.98 | 11.88 | 12.41 | 11.43 |
| | LAR | 27.02 | 56.67 | 68.91 | 88.18 | 60.20 | 7.56 | 7.68 | 8.45 | 8.94 | 8.16 |
| | GCD + LLM TR | 29.43 | 60.18 | 73.21 | 91.32 | 63.54 | 11.07 | 11.35 | 12.59 | 13.17 | 12.05 |
| EfficientViT M2 | GCD + LAR | 31.26 | 62.31 | 75.70 | 92.38 | 65.41 | 12.24 | 12.75 | 13.80 | 14.27 | 13.27 |
| | GCD | 26.87 | 55.58 | 69.00 | 89.38 | 60.01 | 9.48 | 9.84 | 10.95 | 11.38 | 10.41 |
| | LAR | 24.35 | 50.49 | 63.26 | 84.00 | 55.53 | 5.75 | 6.00 | 6.98 | 7.26 | 6.50 |
| | GCD + LLM TR | 27.61 | 56.49 | 70.03 | 89.76 | 60.97 | 10.18 | 10.69 | 11.48 | 12.04 | 11.10 |

Table 8: Results of different losses on CIRR and CIRCO validation set. GCD: global contrastive distillation; LAR: local alignment regularization; LLM TR: text regularization with the sentences made by LLM. The best and second-best results are highlighted in bold and underlined fonts, respectively.

## 5 CONCLUSION

In this paper, we focus on a new but more practical scenarios of asymmetric zero-shot composed image retrieval, which eliminates the requirement of meticulously-labeled training triplets and is flexible for deployment on a resource-constrained platform. Specifically, lightweight model is adopted for the query side while large visual encoder of foundation model is adopted for the gallery side. Adaptive token learner is proposed as the semantic filter to automatically select discriminative semantic groups from image and map them to a sentence, maintaining the fidelity of visual semantics within the word embedding space. By directly concatenating the mapped sentence with text modifier, we are able to compose the multi-modal query to achieve composed retrieval. Global contrastive distillation is applied to align the light model with large foundation model, and local alignment regularization is further utilized to help the mapped sentence better align with the real caption that faithfully describes the image. Extensive experimental results demonstrate that our framework could both achieve better performance and improve retrieval efficiency with deployment flexibility.

## 6 ACKNOWLEDGEMENT

This work was supported by the National Natural Science Foundation of China under Contract 62102128 and 62021001. It was also supported by the GPU cluster built by MCC Lab of Information Science and Technology Institution of USTC.

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

## A APPENDIX

In the appendix, we first present detailed descriptions for the three evaluation datasets.

FashionIQ Wu et al. (2021) is a fashion domain dataset, and it includes 77684 fashion images which are divided into three different categories: Dress, Toptee, and Shirt. The training set comprises 18000 triplets, and in total 46609 images. The validation set includes 15537 images and 6017 triplets. Each query image in the validation set has only one target image. It also has a test set but without ground-truth labels, thus the test set is not used for evaluation.

CIRR Liu et al. (2021) is a dataset of natural domain, and includes 21552 real-life images. It involves 36554 triplets, and are randomly assigned in 80% for training, 10% for validation and 10% for test. Each query image in the validation and test set has only one target image.

CIRCO Baldrati et al. (2023a) is collected from MSCOCO and consists of a validation set and a test set, which includes 220 and 800 query images, respectively. Each query image has a text modifier and more than one target images. The database involves 120k images.

We report the comparison with the state-of-the-art methods on CIRR and CIRCO validation sets in Table 9 and 10, respectively. Accompanied with Table 2 and 3, our method demonstrates consistent superiority over the simple baselines (Image-only, Text-only and Image + Text) and the state-of-the-art methods of Pic2word and SEARLE.

| Gallery model | Query model | Method | Recall@K | | | | Average |
| | | | R@1 | R@5 | R@10 | R@50 | |
|---|---|---|---|---|---|---|---|
| CLIP ViT-L/14 | – | Image-only | 7.06 | 25.52 | 35.64 | 60.18 | 32.10 |
| | – | Text-only | 21.19 | 46.14 | 57.43 | 78.74 | 50.88 |
| | – | Image + Text | 10.79 | 34.37 | 47.05 | 75.27 | 41.87 |
| | – | Pic2Word | 22.54 | 51.69 | 64.76 | 86.19 | 56.30 |
| | – | SEARLE | 25.07 | 55.20 | 68.52 | 90.24 | 59.76 |
| BLIP | – | Image-only | 6.72 | 23.13 | 32.82 | 55.66 | 29.58 |
| | – | Text-only | 25.38 | 51.02 | 61.76 | 81.68 | 54.96 |
| | – | Image + Text | 7.58 | 25.90 | 36.28 | 61.13 | 32.72 |
| | – | Pic2word | 27.89 | 58.15 | 69.42 | 87.53 | 60.75 |
| | – | SEARLE | 28.03 | 60.74 | 71.41 | 88.82 | 62.25 |
| | – | Ours(Sym) | 30.21 | 61.52 | 74.09 | 90.81 | 64.16 |
| | EfficientNet B2 | Ours(Asy) | 31.48 | 63.42 | **77.27** | **93.16** | **66.33** |
| | EfficientViT M2 | Ours(Asy) | 31.26 | 62.31 | 75.70 | 92.38 | 65.41 |

Table 9: Results of zero-shot composed image retrieval on CIRR validation set.

| Gallery model | Query model | Method | mAP@K | | | | Average |
| | | | mAP@5 | mAP@10 | mAP@25 | mAP@50 | |
|---|---|---|---|---|---|---|---|
| CLIP ViT-L/14 | – | Image-only | 1.85 | 2.24 | 3.11 | 3.57 | 2.69 |
| | – | Text-only | 2.78 | 3.06 | 3.72 | 3.95 | 3.38 |
| | – | Image + Text | 4.71 | 5.74 | 6.88 | 7.44 | 6.19 |
| | – | Pic2Word | 7.50 | 8.31 | 9.45 | 10.11 | 8.84 |
| | – | SEARLE | 11.02 | 12.12 | 13.62 | 14.43 | 12.80 |
| BLIP | – | Image-only | 1.86 | 2.18 | 2.71 | 2.97 | 2.43 |
| | – | Text-only | 5.32 | 5.76 | 6.04 | 6.29 | 5.85 |
| | – | Image + Text | 2.04 | 2.63 | 3.18 | 3.49 | 2.84 |
| | – | Pic2Word | 7.91 | 8.27 | 9.05 | 9.56 | 8.70 |
| | – | SEARLE | 11.52 | 11.74 | 12.91 | 13.56 | 12.43 |
| | – | Ours(Sym) | 11.03 | 12.01 | 13.17 | 13.81 | 12.51 |
| | EfficientNet B2 | Ours(Asy) | **13.19** | **13.83** | **15.20** | **15.85** | **14.52** |
| | EfficientViT M2 | Ours(Asy) | 12.24 | 12.75 | 13.80 | 14.27 | 13.27 |

Table 10: Results of zero-shot composed image retrieval on CIRCO validation set.

We also report the results of ablation study on FashionIQ dataset.

Table 11 shows adopting different mapping networks during training, which demonstrates consistent superiority for both EfficientNet B2 and EfficientViT M2.

Table 12 shows utilizing different query models for ZSCIR, which shows the superiority of lightweight models with pre-training initialization.

Table 13 shows the comparison of symmetric and asymmetric retrieval on FashionIQ dataset, and shows consistent superiority of our asymmetric structure.

| Query model | Mapping network | Dress | | Shirt | | Top&tee | | Average | |
|---|---|---|---|---|---|---|---|---|---|
| | | R@10 | R@50 | R@10 | R@50 | R@10 | R@50 | R@10 | R@50 |
| EfficientNet B2 | transformer | 25.33 | **46.26** | **30.03** | 48.58 | **33.45** | 53.80 | **29.60** | **49.54** |
| EfficientNet B2 | MLP (single token) | 23.65 | 43.83 | 28.46 | 47.74 | 31.16 | 51.96 | 27.76 | 47.84 |
| EfficientViT M2 | transformer | **25.48** | 45.51 | 29.64 | **48.68** | 32.94 | **54.31** | 29.35 | 49.50 |
| EfficientViT M2 | MLP (single token) | 24.39 | 44.27 | 29.05 | 47.45 | 30.75 | 53.19 | 28.06 | 48.30 |

Table 11: Results of different mapping networks with BLIP on FashionIQ validation set. The best results are shown in bold fonts.

| Query model | Dress | | Shirt | | Top&tee | | Average | |
|---|---|---|---|---|---|---|---|---|
| | R@10 | R@50 | R@10 | R@50 | R@10 | R@50 | R@10 | R@50 |
| EfficientNet B0 | 25.19 | 45.56 | 28.95 | 48.04 | 32.53 | 52.78 | 28.89 | 48.79 |
| EfficientNet B0* | 23.05 | 44.17 | 27.43 | 45.14 | 29.98 | 51.76 | 26.82 | 47.02 |
| EfficientNet B2 | 25.33 | **46.26** | **30.03** | 48.58 | **33.45** | 53.80 | **29.60** | **49.54** |
| EfficientNet B2* | 25.04 | 44.22 | 28.31 | 46.76 | 31.26 | 53.24 | 28.20 | 48.07 |
| EfficientViT M2 | **25.48** | 45.51 | 29.64 | **48.68** | 32.94 | **54.31** | 29.35 | 49.50 |
| EfficientViT M2* | 24.14 | 43.63 | 27.63 | 45.58 | 31.72 | 53.24 | 27.83 | 47.48 |
| MobileNet V2 | 24.34 | 45.51 | 28.95 | 47.35 | 32.02 | 54.11 | 28.44 | 48.99 |
| MobileNet V2* | 24.59 | 44.47 | 27.48 | 45.78 | 31.51 | 53.19 | 27.86 | 47.81 |
| MobileViT V2 | 22.16 | 40.46 | 26.35 | 43.23 | 27.84 | 49.01 | 25.45 | 44.23 |
| MobileViT V2* | 21.52 | 39.41 | 25.66 | 42.89 | 27.54 | 48.70 | 24.91 | 43.67 |

Table 12: Results of different query models with BLIP on FashionIQ validation set. * denotes adopting random initialization for light query model. The best results are shown in bold fonts.

Table 14 demonstrates the influence of different losses on FashionIQ, which further verifies the effectiveness of GCD and LAR losses.

To better understand the performance of zero-shot methods, we compare our methods with the supervised methods that adopt the labeled-triplets for training, including CIRPLANT Liu et al. (2021), ARTEMIS Delmas et al. (2021), and CLIP4Cir Baldrati et al. (2022b). We report the results on CIRR and fashionIQ, Since CIRCO does not include training set so that there is no supervised results for it. From Table 15 and 16, our method could surpass CIRPLANT and ALTEMIS, which shows the effectiveness of leveraging pre-trained BLIP model for zero-shot learning. Although there is still gap compared with CLIP4Cir, our method is getting closer with the supervised state-of-the-art method in contrast to previous zero-shot methods, while achieving deployment flexibility with the asymmetric framework.

We visualize the results of the attention map of sentence tokens with the text modifier, and to compare them with the spatial attention of adaptive token learner together with the retrieval results. Some good retrieval results are shown in Figure 5. We could see that there are some tokens that attend to both concepts of query images and concepts of text modifier. In the up-right example, the third and

| Query model | Gallery model | Dress | | Shirt | | Toptee | | Average | |
|---|---|---|---|---|---|---|---|---|---|
| | | R@10 | R@50 | R@10 | R@50 | R@10 | R@50 | R@10 | R@50 |
| EfficientNet B2 | BLIP VE | 25.33 | **46.26** | 30.03 | 48.58 | 33.45 | 53.80 | 29.60 | 49.54 |
| EfficientNet B2 | BLIP VE | 0.5 | 2.08 | 0.64 | 2.01 | 0.87 | 1.63 | 0.67 | 1.91 |
| EfficientViT M2 | BLIP VE | **25.48** | 45.51 | 29.64 | 48.68 | 32.94 | 54.31 | 29.35 | 49.50 |
| EfficientViT M2 | BLIP VE | 0.69 | 1.59 | 1.08 | 2.16 | 1.22 | 2.04 | 1.00 | 1.93 |
| BLIP VE † | BLIP VE | 24.89 | 45.36 | **30.18** | **49.07** | **35.65** | **55.12** | **30.24** | **49.85** |
| BLIP VE | BLIP VE | 2.13 | 5.7 | 1.86 | 4.47 | 2.4 | 6.12 | 2.13 | 5.43 |

Table 13: Comparison with symmetric retrieval on the validation set of FashionIQ. BLIP VE denotes the BLIP visual encoder. † denotes updating the query BLIP VE while preserving the pre-trained BLIP VE in gallery side during training for maintaining the multi-modality alignment. The best results are shown in bold fonts..

| Query model | Variants | Dress | | Shirt | | Top&tee | | Average | |
|---|---|---|---|---|---|---|---|---|---|
| | | R@10 | R@50 | R@10 | R@50 | R@10 | R@50 | R@10 | R@50 |
| EfficientNet B2 | GCD + LAR | 25.33 | **46.26** | **30.03** | 48.58 | **33.45** | 53.80 | **29.60** | **49.54** |
| | GCD | 23.90 | 45.66 | 27.33 | 45.14 | 31.31 | 52.12 | 27.51 | 47.64 |
| | LAR | 22.76 | 42.69 | 26.15 | 44.16 | 29.68 | 50.59 | 26.20 | 45.81 |
| | GCD + LLM TR | 24.19 | 45.17 | 27.92 | 44.90 | 31.06 | 51.56 | 27.72 | 47.21 |
| EfficientViT M2 | GCD + LAR | **25.48** | 45.51 | 29.64 | **48.68** | 32.94 | **54.31** | 29.35 | 49.50 |
| | GCD | 23.35 | 44.03 | 26.79 | 44.80 | 30.65 | 50.99 | 26.93 | 46.61 |
| | LAR | 13.73 | 28.46 | 21.49 | 38.13 | 24.68 | 43.50 | 19.97 | 36.70 |
| | GCD + LLM TR | 23.80 | 44.03 | 26.94 | 45.00 | 30.49 | 50.94 | 27.08 | 46.66 |

Table 14: Results of different losses on FashionIQ validation set. GCD: global contrastive distillation; LAR: local alignment regularization; LLM TR: text regularization that adopts the sentences made by LLM. The best and second-best results are highlighted in bold and underlined fonts, respectively.

| Supervision | Method | Recall@K | | | | Avg |
|---|---|---|---|---|---|---|
| | | R@1 | R@5 | R@10 | R@50 | |
| Supervised | CIRPLANT | 19.60 | 52.60 | 68.40 | 92.40 | 58.25 |
| | ALTEMIS | 16.96 | 46.10 | 61.31 | 87.73 | 53.03 |
| | CLIP4Cir | **33.60** | **65.40** | **77.40** | **95.20** | **67.90** |
| Zero-shot | Pic2Word(CLIP) | 23.90 | 51.70 | 65.30 | 87.80 | 57.17 |
| | SEARLE(CLIP) | 24.22 | 52.41 | 66.29 | 88.63 | 57.88 |
| | Pic2word(BLIP) | 26.70 | 53.16 | 64.10 | 84.36 | 57.08 |
| | SEARLE(BLIP) | 29.27 | 54.86 | 66.57 | 86.16 | 59.21 |
| | Ours(Asy, EN B2) | 30.84 | 61.06 | 73.57 | 92.43 | 64.48 |
| | Ours(Asy, EVT M2) | 29.63 | 58.99 | 71.37 | 91.47 | 62.87 |

Table 15: Comparsion with supervised methods on CIRR test set. The best and second best results are shown in bold and underline fonts, respectively.

the 6-th tokens focus on "cat" and "fruits" respectively, and the 4-th and 6-th are paying attention to the concepts in the text modifier, *i.e.*, "vegetables" and "bench", respectively. These examples could extract distinctive visual concepts and interact with the text modifier correspondingly, thus achieving good retrieval performance.

We also present some failure cases in Figure 6. The failure retrieval could be attributed to the wrong associated concept, rare words, and abstract concepts. For the example in the middle, although the 4-th token catches the concept of "woman" and attends to the "phone", there is difficulty associating "them" in the text modifier with "dog", *i.e.*, the 6-th attends to "them" but fail to catch the concept

| Supervision | Method | Dress | | Shirt | | Top&tee | | Average | |
|---|---|---|---|---|---|---|---|---|---|
| | | R@10 | R@50 | R@10 | R@50 | R@10 | R@50 | R@10 | R@50 |
| Supervised | CIRPLANT | 17.45 | 40.41 | 17.53 | 38.81 | 21.64 | 45.38 | 18.87 | 41.53 |
| | ALTEMIS | 27.20 | 52.40 | 21.80 | 43.60 | 29.20 | 54.80 | 26.10 | 50.30 |
| | CLIP4Cir | **30.30** | **54.50** | **37.20** | **55.80** | **39.20** | **61.30** | **35.60** | **57.20** |
| Zero-shot | Pic2Word(CLIP) | 20.00 | 40.20 | 26.20 | 43.60 | 27.90 | 47.40 | 24.70 | 43.70 |
| | SEARLE(CLIP) | 20.32 | 43.18 | 27.43 | 45.68 | 29.32 | 50.17 | 25.69 | 46.34 |
| | Pic2word(BLIP) | 21.35 | 42.68 | 27.51 | 46.01 | 29.12 | 49.33 | 25.99 | 46.00 |
| | SEARLE(BLIP) | 22.11 | 41.79 | 29.72 | 48.53 | 31.03 | 52.37 | 27.62 | 47.56 |
| | Ours(Asy, EN B2) | 25.33 | 46.26 | 30.03 | 48.58 | 33.45 | 53.80 | 29.60 | 49.54 |
| | Ours(Asy, EVT M2) | 25.48 | 45.51 | 29.64 | 48.68 | 32.94 | 54.31 | 29.35 | 49.50 |

Table 16: Comparsion with supervised methods on FashionIQ validation set. The best and second best results are shown in bold and underline fonts, respectively.

of "dog" in the visual feature map. For the example in the bottom, it is difficult for the model to grasp the abstract concept "pattern" and "less of a V-neck", thus leading to wrong retrieval results.

We present more visualization comparison of different token lengths in Figure 7 and 8. Generally, to achieve accurate retrieval, the pseudo sentence tokens need to accomplish two aspects: extracting effective visual information and interacting with textual information. When the token length is small, there would be drop of performance due to the lack of capacity to extract sufficient visual information. As shown in Figure 7, query model with small token length performs worse even if the attention maps are reasonable. When the token length increases for too long, on one hand, some tokens may focus on trivial and noisy visual patterns, as shown in Figure 4; on the other hand, these tokens may interact incorrectly with the text modifier, and may impact the correct token-text interaction. As shown in Figure 8, the excessive tokens associate the visual pattern with incorrect text information, potentially interfering with the interaction of multi-modal information. Even if there are accurate associations between other sentence tokens and text modifier, this interference would still degrade the retrieval accuracy.

To verify the efficiency of our framework, we report the latency on both query side and cloud side on CIRCO vailidation set as shown in Table 17. For query side, we report the inference latency for different query models. We find that lightweight encoders are generally twice faster than large visual encoder (BLIP VE). For cloud side, since the gallery models are the same (BLIP TE & VE), the retrieval latency is the same.

To further verify the effectiveness of adaptive token learner, we carry out experiments for other three variants of mapping networks:

(I) simple tokenizer (ST): only using the block described in Eq. (1) and Eq. (2) in section 3.1 in the manuscript.

(II) adaptive token learner without cross-attention (ATL w/o CA): including the blocks described in Eq. (1), (2), (3).

(III) adaptive token learner without cross-attention (ATL w/o SA): including the blocks described in Eq. (1), (2), (4).

We report the results on EfficientNet B2 as below. From the table, it could be seen that with more components added to the mapping network, the performance gradually improves on three benchmark datasets. The simple tokenizer divides the semantics in different groups, which preserves more visual information than MLP. The self-attention models the mutual relationship between different visual tokens and cross-attention filters out noisy information, enhancing the pseudo sentence tokens to be more robust and refined visual representations in word embedding space. Therefore, the full version of adaptive token learner (ATL) achieves the best performance.

| Query model | Query inference latency (ms, per query) | Gallery model | Retrieval latency (ms, per query) | Total retrieval latency |
|---|---|---|---|---|
| EfficientNet B0 | 3.07 | BLIP VE | 3.58 | 6.65 |
| EfficientNet B2 | 3.24 | BLIP VE | 3.58 | 6.82 |
| EfficientViT M2 | 3.27 | BLIP VE | 3.58 | 6.85 |
| MobileNet V2 | 3.03 | BLIP VE | 3.58 | 6.61 |
| MobileViT V2 | 3.40 | BLIP VE | 3.58 | 6.98 |
| BLIP VE | 6.53 | BLIP VE | 3.58 | 10.11 |
| CLIP ViT-L/14 | 24.44 | CLIP ViT-L/14 | 3.60 | 15.38 |

Table 17: Inference latency on query side and gallery side for different query and gallery models.

| Mapping network | Components | | | Recall@K | | | | Avg |
|---|---|---|---|---|---|---|---|---|
| | ST | SA | CA | R@1 | R@5 | R@10 | R@50 | |
| MLP | | | | 30.18 | 59.46 | 71.30 | 89.33 | 62.57 |
| ST | ✓ | | | 29.94 | 61.68 | 73.71 | 91.92 | 64.31 |
| ATL w/o CA | ✓ | ✓ | | 29.85 | 61.61 | 74.48 | 91.53 | 64.37 |
| ATL w/o SA | ✓ | | ✓ | 30.35 | 62.13 | 75.95 | 91.89 | 65.08 |
| ATL | ✓ | ✓ | ✓ | **31.48** | **63.42** | **77.27** | **93.16** | **66.33** |

Table 18: Ablation study on variants of mapping networks on CIRR validation set. Best results are shown in bold fonts.

| Mapping network | Components | | | mAP@K | | | | Avg |
|---|---|---|---|---|---|---|---|---|
| | ST | SA | CA | mAP@1 | mAP@5 | mAP@10 | mAP@50 | |
| MLP | | | | 8.28 | 8.67 | 9.62 | 10.00 | 9.14 |
| ST | ✓ | | | 9.89 | 10.33 | 11.38 | 11.79 | 10.85 |
| ATL w/o CA | ✓ | ✓ | | 10.46 | 10.94 | 12.03 | 12.52 | 11.49 |
| ATL w/o SA | ✓ | | ✓ | 11.52 | 12.15 | 13.83 | 13.58 | 12.77 |
| ATL | ✓ | ✓ | ✓ | **13.19** | **13.83** | **15.20** | **15.85** | **14.52** |

Table 19: Ablation study on variants of mapping networks on CIRCO validation set. Best results are shown in bold fonts.

| Mapping network | Components | | | Dress | | Shirt | | Top&tee | | Average | |
|---|---|---|---|---|---|---|---|---|---|---|---|
| | ST | SA | CA | R@10 | R@50 | R@10 | R@50 | R@10 | R@50 | R@10 | R@50 |
| MLP | | | | 23.65 | 43.83 | 28.46 | 47.74 | 31.16 | 51.96 | 27.76 | 47.84 |
| ST | ✓ | | | 23.45 | 43.58 | 28.80 | 45.78 | 31.57 | 52.32 | 27.94 | 47.23 |
| ATL w/o CA | ✓ | ✓ | | 24.00 | 45.86 | 27.33 | 45.29 | 31.62 | 52.27 | 27.65 | 47.80 |
| ATL w/o SA | ✓ | | ✓ | 24.15 | **46.71** | 28.16 | 46.22 | 32.46 | 52.37 | 28.26 | 48.43 |
| ATL | ✓ | ✓ | ✓ | **25.33** | 46.26 | **30.03** | **48.58** | **33.45** | **53.80** | **29.60** | **49.54** |

Table 20: Ablation study on variants of mapping networks on FashionIQ. Best results are shown in bold fonts.

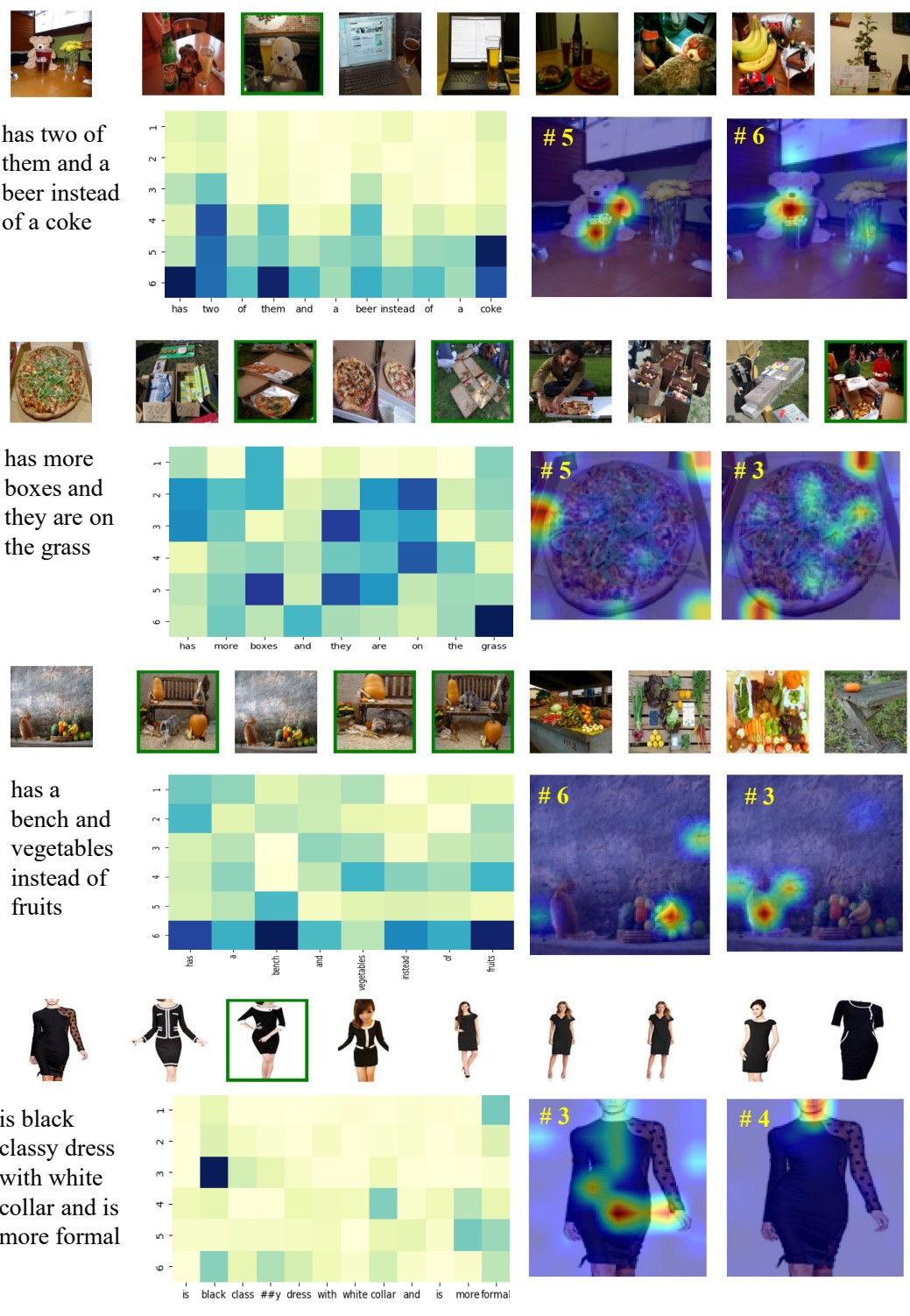

Figure 5: Some good retrieval examples and the attention map of sentence tokens on visual feature map and with modifier text. Ground-truth target images are marked in green boxes.

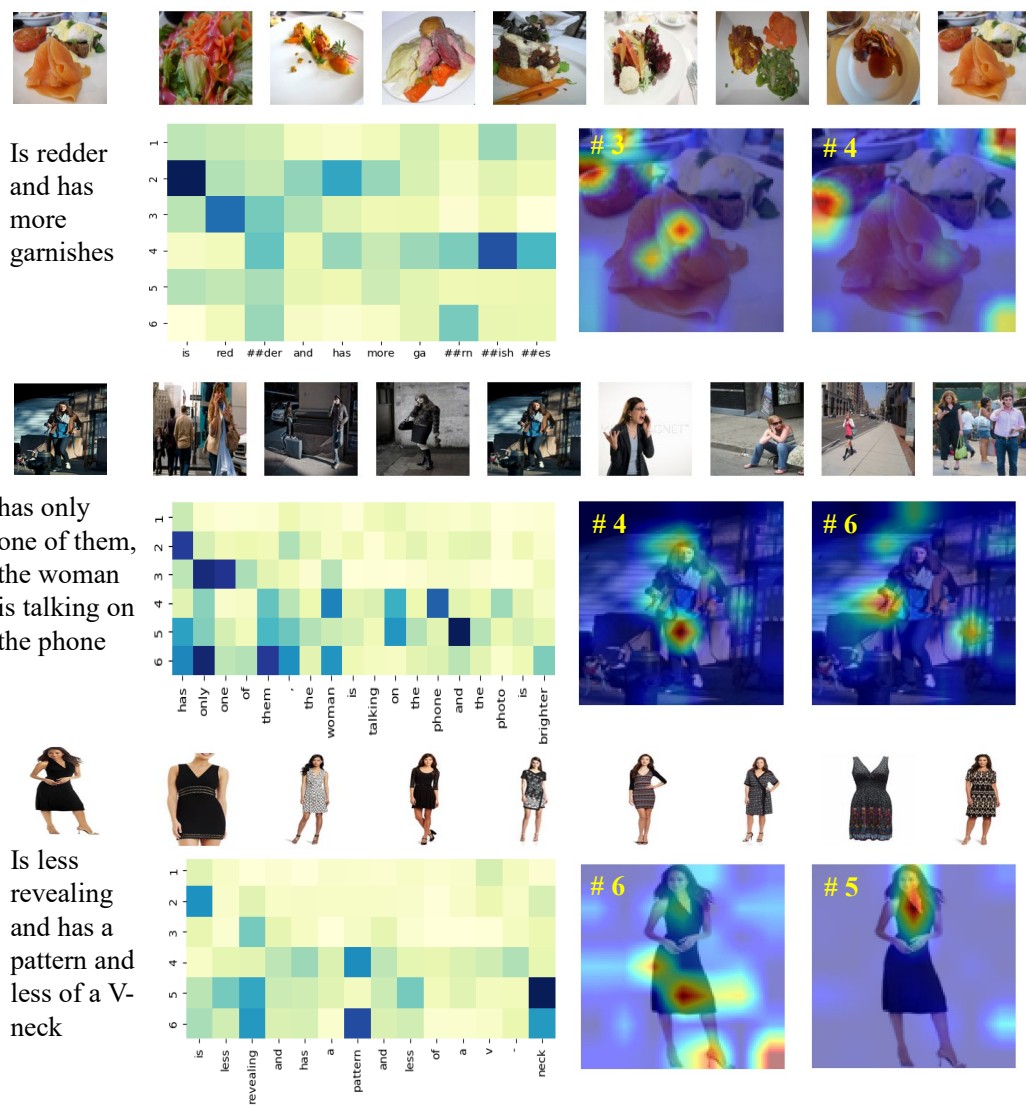

Figure 6: Some failure retrieval case and the attention map of sentence tokens on visual feature map and with modifier text. Ground-truth target images are marked in green boxes.

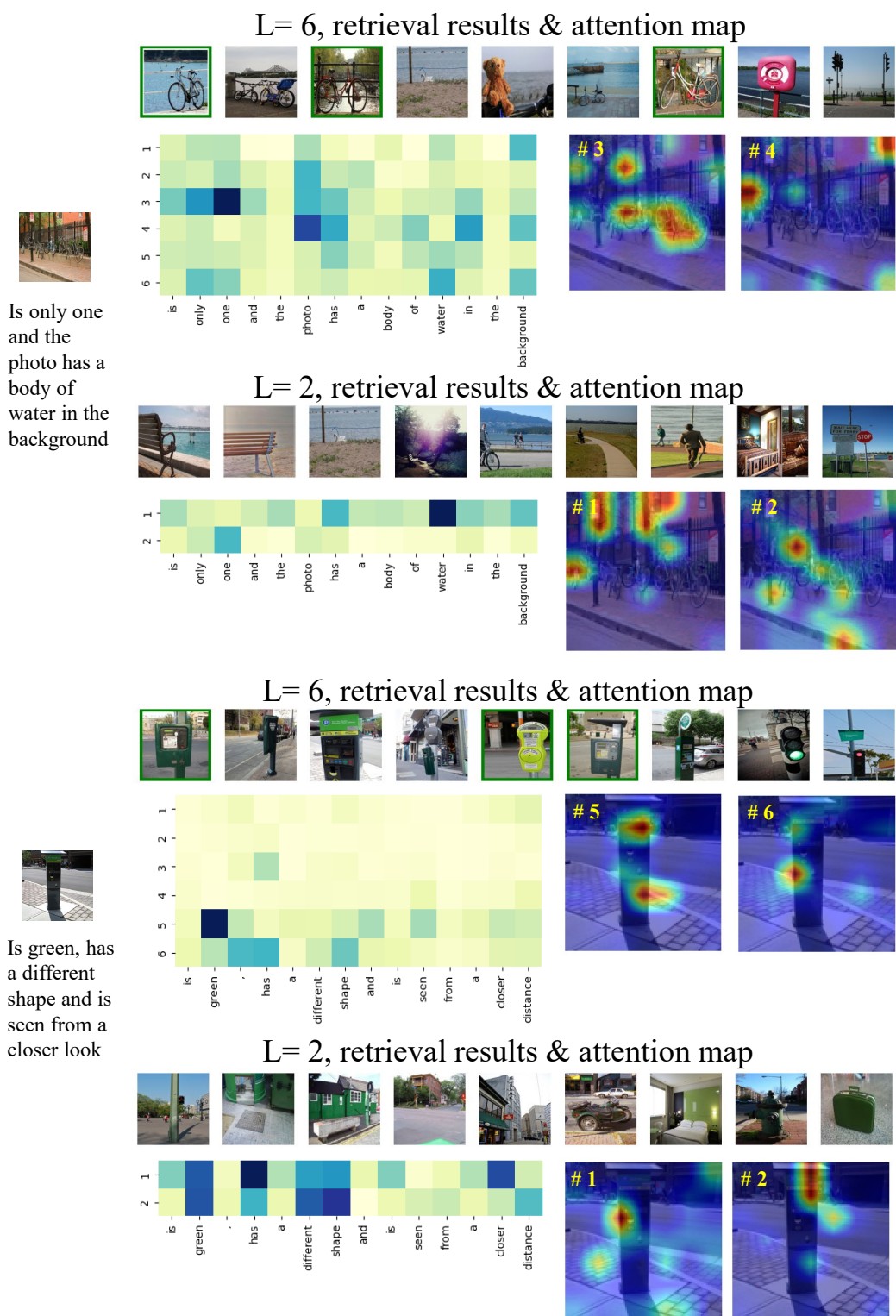

Figure 7: More comparison results of different token lengths (2 v.s. 6). Query image and text modifier are shown on the left, and the attention map of sentence tokens with text modifiers and on visual feature map are shown on the right. Ground-truth target images are marked in green boxes.

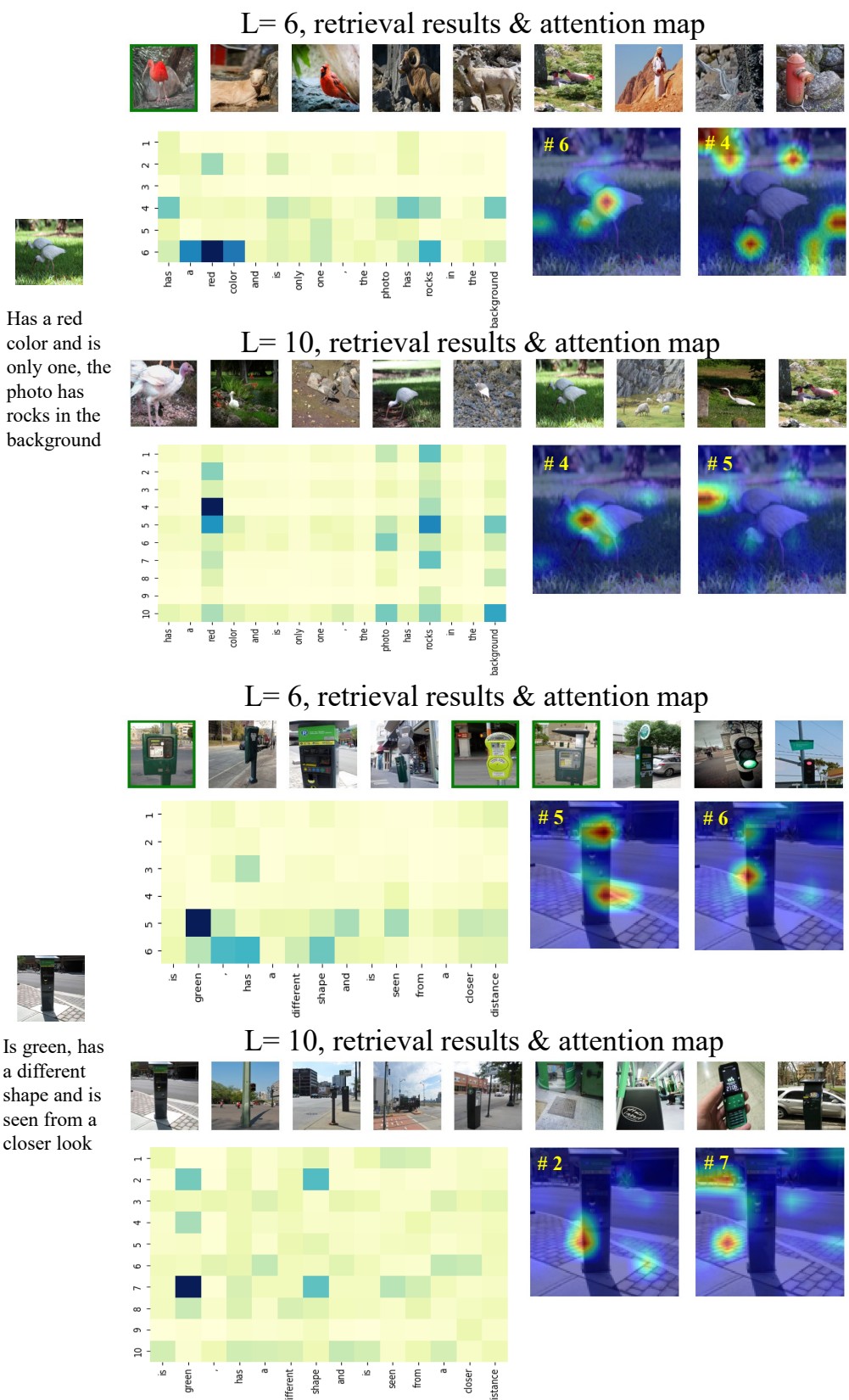

Figure 8: More comparison results of different token lengths (6 v.s. 10). Query image and text modifier are shown on the left, and the attention map of sentence tokens with text modifiers and on visual feature map are shown on the right. Ground-truth target images are marked in green boxes.

