# OpenReview forum: "Image2Sentence based Asymmetrical Zero-shot Composed Image Retrieval"
_ICLR.cc/2024/Conference — ICLR 2024 spotlight_

### Official Review · Reviewer_zR31 · 2023-10-23

**Soundness:** 3 good
**Presentation:** 3 good
**Contribution:** 3 good
**Rating:** 8
**Confidence:** 5

**Summary:**

This paper aims at the task of composed image retrieval (CIR) which retrieves images by providing a multimodal query such as a query image and additional text which describes the user's further query intention. Usually, such task is resolved by aligning the multimodal query and gallery features with vision-and-language pretraining and finetuning techniques. The authors argue that existing methods are not feasible to mobile applications due to expensive computational costs by forwarding large multimodal foundation models on mobile devices. The proposed solution is adopting a lightweight model to process the query while still maintaining the large foundation model for the gallery side. In order to bridge the representation gap between the lightweight and large model, the adaptive token learner is proposed to map an image to a sentence in the language model space. Finally, the authors verify their contributions on three evaluation benchmarks.

**Strengths:**

[1] Overall, this paper is well written and easy to follow.

[2] I like the motivation of this work since deploying original large foundation model is almost impossible in mobile applications. Different from pruning or distilling such heavy models, the authors proposed the lightweight encoder with a tunable adaptive token learner. The idea behind is borrowed from the LLM-Adapters.

[3] The proposed modules are technically sound and the experimental resutls are sufficient. The training resources are extremly friendly with 4 RTX 3090 GPUs.

**Weaknesses:**

[1] Since this work is a retrieval task, it is important to report how the retrieval performance varies as the gallery size scales up. I understand that the three evalutation datasets are standard benchmarks. However, it would make the contribution more solid if millions distractors could be involved in the gallery, although most existing SOTAs didn't report such results.

[2] The inference time should be reported including in the query side and in the cloud side.

[3] Some minor issues are listed in the next part.

**Questions:**

[1] Figure 1 could be further improved. Specifically, the pink rectangle and trapezoid could be shrunk and the blue trapezoids could be enlarged. As a result, it is much easier to quickly grasp the idea at first glance for readers.

[2] Table 5 provides a viriant mapping network which is actually a MLP. Are there other options?

[3] Figure 3 reveals a fact that larger token lengths could degrade the retrieval performance. The authors attribute such fact to the background noise or the trivial patterns. Are there any deeper insights or visualization analysis?

---

> ### Author Response · Authors · 2023-11-21
> **To Reviewer zR31**
>
> We appreciate the detailed comments and acknowledgment of our contributions. We provide the responses as follows.
>
> **Q1**: Since this work is a retrieval task, it is important to report how the retrieval performance varies as the gallery size scales up. I understand that the three evalutation datasets are standard benchmarks. However, it would make the contribution more solid if millions distractors could be involved in the gallery, although most existing SOTAs didn't report such results.
>
> **R1**: We adopt the distractors from R1M in [4] to scale up the gallery set during evaluation. Note that for CIRR and CIRCO, the evaluation on test set could only be conducted on the official cloud server, thus we add distractors to gallery for the validation set. The results on three benchmarks are shown as below, which reveal that the distractors in the gallery would dampen the retrieval accuracy, however our method is still superior on three benchmarks.
>
> #### Scaling up the gallery of FashionIQ validation set
> Method | Scale of distractors | Dress R@10 | Dress R@50 | Shirt R@10 | Shirt R@50 | Toptee R@10 | Toptee R@50 | R@10 Avg | R@50 Avg |
> | :------: | :------:| :------:| :------: | :------: | :------: | :------: | :------: | :------: | :------: |
> | Pic2word | 0 | 21.35 | 42.68 | 27.51 | 46.01 | 29.12 | 49.33 | 25.99 | 46.00 |
> | Pic2word | 1M | 19.88 | 39.94 | 26.30 | 44.99 | 27.40 | 47.73 | 24.53 | 44.22 |
> | SEARLE | 0 | 22.11 | 41.79 | 29.72 | 48.53 | 31.03 | 52.37 | 27.62 | 47.56 |
> | SEARLE | 1M | 21.24 | 39.77 | 28.95 | 47.79 | 30.90 | 50.95 | 27.03 | 46.17 |
> | Ours | 0 | 25.33 | 46.26 | 30.03 | 48.58 | 33.45 | 53.80 | 29.60 | 49.54 |
> | Ours | 1M | 25.24 | 45.46 |  29.54 | 47.30 | 32.18 | 52.32 | 28.99 | 48.36 |
>
> #### Scaling up the gallery of CIRR validation set
>
> | Method | Scale of distractors | R@1 | R@5 | R@10 | R@50 | Avg |
> | :------: | :------:| :------:| :------: | :------: | :------: | :------: |
> | Pic2word | 0 |  27.89 | 58.15 | 69.42 | 87.53 | 60.75 |
> | Pic2word | 1M | 19.47 | 38.01 | 46.76 | 64.15 | 42.10 |
> | SEARLE | 0 | 28.03 | 60.74 | 71.41 | 88.82 | 62.25 |
> | SEARLE | 1M | 21.84 | 43.70 | 53.55 | 70.96 | 47.51
> | Ours | 0 | 31.48 | 63.42 | 77.27 | 93.16 | 66.33 |
> | Ours | 1M | 22.77 | 45.64 | 55.49 | 72.26 | 49.04 |
>
>
> #### Scaling up the gallery of CIRCO validation set
>
> | Method | Scale of distractors | mAP@5 | mAP@10 | mAP@25 | mAP@50 | Avg |
> | :------: | :------:| :------:| :------: | :------: | :------: | :------: |
> | Pic2word | 0 | 7.91 | 8.27 | 9.05 | 9.56 | 8.70 |
> | Pic2word | 1M | 4.85 | 5.12 | 5.58 | 5.81 | 5.34 |
> | SEARLE | 0 |  11.52 | 11.74 | 12.91 | 13.56 | 12.43 |
> | SEARLE | 1M | 7.14 | 7.55 | 8.15 | 8.47 | 7.83 |
> | Ours | 0 | 13.19 | 13.83 | 15.20 | 15.85 | 14.52 |
> | Ours | 1M | 8.66 | 9.16 | 10.20 | 10.61 | 9.66 |
>
> [4] Radenović, F., Iscen, A., Tolias, G., Avrithis, Y., & Chum, O. (2018). Revisiting oxford and paris: Large-scale image retrieval benchmarking. In Proceedings of the IEEE conference on computer vision and pattern recognition (pp. 5706-5715).
>
> **Q2**: The inference time should be reported including in the query side and in the cloud side.
>
> **R2**: We report the latency on both query side and cloud side on CIRCO vailidation set. For query side, we report the inference latency for different query models. We find that lightweight encoders are generally twice faster than large visual encoder (BLIP VE). For cloud side, since the gallery models are the same (BLIP TE & VE), the retrieval latency is the same.
>
> | Query model | Query inference latency (ms, per query) | Gallery model | Retrieval latency (ms, per query) | Total retrieval latency (ms, per query) |
> | :------:| :------:| :------: | :------: | :------: |
> | EfficientNet B0 | 3.07 | BLIP VE | 3.58 | 6.65 |
> | EfficientNet B2 | 3.24 | BLIP VE | 3.58 | 6.82 |
> | EfficientViT M2 | 3.27 | BLIP VE | 3.58 | 6.85 |
> | MobileNet V2 | 3.03 | BLIP VE | 3.58 | 6.61 |
> | MobileViT V2 | 3.40 | BLIP VE | 3.58 | 6.98 |
> | BLIP VE | 6.53 | BLIP VE | 3.58 | 10.11 |
>
> **Q3**: Figure 1 could be further improved. Specifically, the pink rectangle and trapezoid could be shrunk and the blue trapezoids could be enlarged. As a result, it is much easier to quickly grasp the idea at first glance for readers.
>
> **R3**: Thanks for the suggestions of Figure 1. We have revised Figure 1 for better clarity.

---

> > ### Author Response · Authors · 2023-11-21
> > **To Reviewer zR31**
> >
> > **Q4**: Table 5 provides a viriant mapping network which is actually a MLP. Are there other options?
> >
> > **R4**: There are other three variants of mapping networks:
> >
> > (I) simple tokenizer (ST): only using the block described in Eq.(1) and Eq.(2) in section 3.1 in the manuscript.
> >
> > (II) adaptive token learner without cross-attention (ATL w/o CA): including the blocks described in Eq.(1), (2), (3).
> >
> > (III) adaptive token learner without cross-attention (ATL w/o SA): including the blocks described in Eq.(1), (2), (4).
> >
> > We report the results on EfficientNet B2 as below. From the table, it could be seen that with more components added to the mapping network, the performance gradually improves on three benchmark datasets. The simple tokenizer divides the semantics in different groups, which preserves more visual information than MLP. The self-attention models the mutual relationship between different visual tokens and cross-attention filters out noisy information, enhancing the pseudo sentence tokens to be more robust and refined visual representations in word embedding space. Therefore, the full version of adaptive token learner (ATL) achieves the best performance.
> >
> > #### Ablation study of mapping networks on FashionIQ dataset
> > | Mapping Network | ST | SA | CA | Dress R@10 | Dress R@50 | Shirt R@10 | Shirt R@50 | Toptee R@10 | Toptee R@50 | R@10 | R@50 |
> > | :------:| :------:| :------:| :------:| :------:| :------: | :------: | :------: | :------: | :------: | :------: | :------: |
> > | MLP |  |  |  | 23.65 | 43.83 | 28.46 | 47.74 | 31.16 | 51.96 | 27.76 | 47.84 |
> > | ST | ✔ | | | 23.45 | 43.58 | 28.80 | 45.78 | 31.57 | 52.32 | 27.94 | 47.23 |
> > | ATL w/o CA | ✔ | ✔|  | 24.00 | 45.86 | 27.33 | 45.29 | 31.62 | 52.27 | 27.65 | 47.80 |
> > | ATL w/o SA | ✔ |  | ✔ |  24.15 | 46.71 | 28.16 | 46.22 | 32.46 | 52.37 | 28.26 | 48.43 |
> > | ATL |✔| ✔ | ✔ | 25.33 | 46.26 | 30.03 | 48.58 | 33.45 | 53.80 | 29.60 | 49.54 |
> >
> > #### Ablation study of mapping networks on CIRR validation set
> >
> > | Mapping Network | ST | SA | CA | R@1 | R@5 | R@10 | R@50 | Avg |
> > | :------:| :------:| :------:|:------:| :------:| :------: | :------: | :------: | :------: |
> > | MLP |  |  |  | 30.18 | 59.46 | 71.30 | 89.33 | 62.57 |
> > | ST | ✔ | | | 29.94 | 61.68 | 73.71 | 91.92 | 64.31 |
> > | ATL w/o CA | ✔ | ✔|  | 29.85 | 61.61 | 74.48 | 91.53 | 64.37 |
> > | ATL w/o SA | ✔ |  | ✔ | 30.35 | 62.13 | 75.95 | 91.89 | 65.08 |
> > | ATL |✔| ✔ | ✔ | 31.48 | 63.42 | 77.27 | 93.16 | 66.33 |
> >
> > #### Ablation study of mapping networks on CIRCO validation set
> >
> > | Mapping Network | ST | SA | CA | mAP@5 | mAP@10 | mAP@25 | mAP@50 | Avg |
> > | :------:| :------:| :------:|:------:| :------:| :------: | :------: | :------: | :------: |
> > | MLP |  |  |  | 8.28 | 8.67 | 9.62 | 10.00 | 9.14 |
> > | ST | ✔ | | | 9.89 | 10.33 | 11.38 | 11.79 | 10.85 |
> > | ATL w/o CA | ✔ | ✔|  | 10.46 | 10.94 | 12.03 | 12.52 | 11.49 |
> > | ATL w/o SA | ✔ |  | ✔ | 11.52 | 12.15 | 13.83 | 13.58 | 12.77 |
> > | ATL |✔| ✔ | ✔ | 13.19 | 13.83 | 15.20 | 15.85 | 14.52 |
> >
> > **Q5**: Figure 3 reveals a fact that larger token lengths could degrade the retrieval performance. The authors attribute such fact to the background noise or the trivial patterns. Are there any deeper insights or visualization analysis?
> >
> > **R5**: Generally, to achieve accurate retrieval, the pseudo sentence tokens need to accomplish two aspects: extracting effective visual information and interacting with textual information. When the token length increases for too long, on one hand, some tokens may focus on trivial and noisy visual patterns, as shown in Figure 4; on the other hand, these tokens may interact incorrectly with the text modifier, and may impact the correct token-text interaction. As shown in Figure 9 in the manuscript appendix, the excessive tokens associate the visual patterns with incorrect text information, potentially interfering with the interaction of multi-modal information. Even if there are accurate associations between other sentence tokens and text modifier, this interference would still degrade the retrieval accuracy.
> >
> > Finally, thank you again for your recognition and positive review to our work. We will incorporate your suggestions into our next revision.
> >
> > Note: since OpenReview does not allow image uploading in the comment box, we place the additional figures in the appendix of revised manuscript.

---

> > > ### Comment · Reviewer_zR31 · 2023-11-22
> > > **Response to the commets**
> > >
> > > Aftering reading the author's detailed response, I think my concerns have already been resolved. Thus, I just upgraded my rating score. Besides, I suggest incorporating these additional results and tables into the final supplementary materials.

---

> > > > ### Author Response · Authors · 2023-11-22
> > > > **To Reviewer zR31**
> > > >
> > > > Thanks for your valuable support. We will follow your suggestion and incorporate these additional results and tables into the final supplementary materials

---

### Official Review · Reviewer_K4T5 · 2023-10-29

**Soundness:** 3 good
**Presentation:** 2 fair
**Contribution:** 3 good
**Rating:** 8
**Confidence:** 5

**Summary:**

This paper introduces an asymmetric zero-shot composed image retrieval framework. The asymmetric retrieval pipeline is established using a lightweight model for query images and a large foundation model for gallery images, enabling feature extraction. Composed image retrieval is achieved by concatenating the sentence representation mapped from the image with a text modifier. To align the features extracted by the lightweight model and the large foundation model, two techniques, namely global contrastive distillation and local alignment regularization, are proposed. Extensive experiments and an ablation study conducted on benchmark datasets have demonstrated the effectiveness of the proposed method.

**Strengths:**

1. An asymmetric zero-shot composed image retrieval framework is proposed.
2. Global contrastive distillation and local alignment regularization techniques are proposed to align features from different models.
3. Extensive experiments are conducted.

**Weaknesses:**

1. The clarity of the writing, particularly in the methods section, requires improvement.

2. For image-only retrieval, could you provide results using the DINO-V2 and MoCo-V3 pretrained models? The CLIP model is typically used for content matching between image and text features.

**Questions:**

t-SNE visualizations could be shown to illustrate the differences between the different methods. This would provide a more intuitive understanding of the feature distributions and separability.

---

> ### Author Response · Authors · 2023-11-21
> **To Reviewer K4T5**
>
> We appreciate the detailed comments and acknowledgment of our contributions. We provide the responses as follows.
>
> **Q1**: The clarity of the writing, particularly in the methods section, requires improvement.
>
> **R1**: Thanks for the suggestion, we have carefully revised our manuscript, with changes marked in blue for clarity. The major modifications include:
> - the description of the framework in section 3
> - the merits of adaptive token learner in section 3.1
> - the clarification of inference phase in section 3.3
>
> Please feel free to share if you have further concerns, we would be keen to address them as well.
>
> **Q2**: For image-only retrieval, could you provide results using the DINO-V2 and MoCo-V3 pretrained models? The CLIP model is typically used for content matching between image and text features.
>
> **R2**: We use the pretrained MoCo-V3 models (resnet50, ViT-base) and DINO-V2 models (ViT-base, ViT-large) to test the retrieval performance on three benchmarks. Note that self-supervised pretrained models are designed for visual modality and do not include the text encoder, thus we merely report the image-only retrieval results for them. For comparison, we also report the retrieval performance of our method that integrates both visual and textual information for retrieval.
>
> #### The image-only results on FashionIQ validation set
>
> | Network | Dress R@10 | Dress R@50 | Shirt R@10 | Shirt R@50 | Toptee R@10 | Toptee R@50 | R@10 Avg | R@50 Avg |
> | :------:| :------:| :------: | :------: | :------: | :------: | :------: | :------: | :------: |
> | CLIP VE | 5.40 | 13.90 | 9.90 | 20.80 | 8.30 | 17.70 | 7.90 | 17.50 |
> | BLIP VE | 4.07 | 11.35 | 7.07 | 16.39 | 6.88 | 13.67 | 6.01 | 13.67 |
> | MoCo-V3 resnet50 | 2.88 | 7.49 | 4.71 | 9.42 | 3.77 | 8.67 | 3.79 | 8.53 |
> | MoCo-V3 ViT-base | 2.93 | 8.23 | 4.32 | 9.81 | 3.67 | 8.62 | 3.64 | 8.89 |
> | DINO-V2 ViT-base | 4.31 | 10.66 | 6.92 | 13.05 | 5.10 | 11.73 | 5.44 | 11.81 |
> | DINO-V2 ViT-large | 4.46 | 11.06 | 6.77 | 12.71 | 6.12 | 12.80 | 5.78 | 12.19 |
> | Ours (multi-modal) | 25.33 | 46.26 | 30.03 | 48.58 | 33.45 | 53.80  | 29.60 | 49.54 |
>
>
> #### The image-only results on CIRR test set
>
> | Network | R@1 | R@5 | R@10 | R@50 | Avg |
> | :------:| :------:| :------: | :------: | :------: | :------: |
> | CLIP VE | 7.40 | 23.60 | 34.00 | 57.40 | 30.60 |
> | BLIP VE | 7.52 | 21.83 | 31.21 | 52.94 | 28.37 |
> | MoCo-V3 resnet50 | 8.53 | 30.58 | 43.69 | 73.35 | 39.04 |
> | MoCo-V3 ViT-base | 8.60 | 30.02 | 42.92 | 70.89 | 38.11 |
> | DINO-V2 ViT-base | 7.88 | 27.06 | 39.04 | 66.27 | 35.06 |
> | DINO-V2 ViT-large | 7.78 | 25.95 | 37.04 | 62.55 | 33.33 |
> | Ours (multi-modal) | 30.84 | 61.06 | 73.57 | 92.43 | 64.48 |
>
> #### The image-only results on CIRCO test set
>
> | Network | mAP@5 | mAP@10 | mAP@25 | mAP@50 | Avg |
> | :------:| :------:| :------: | :------: | :------: | :------: |
> | CLIP VE | 1.69 | 2.18 | 2.78 | 3.22 | 2.47 |
> | BLIP VE | 1.28 | 1.48 | 1.88 | 2.13 | 1.69 |
> | MoCo-V3 resnet50 | 0.83 | 1.01 | 1.28 | 1.41 | 1.13 |
> | MoCo-V3 ViT-base | 0.91 | 1.08 | 1.33 | 1.48 | 1.20 |
> | DINO-V2 ViT-base | 2.03 | 2.49 | 3.10 | 3.48 | 2.78 |
> | DINO-V2 ViT-large | 2.00 | 2.40 | 2.98 | 3.37 | 2.69 |
> | Ours (multi-modal) | 11.33 | 12.25 | 13.42 | 13.97 | 12.74 |
>
> Generally, self-supervised models could improve the performance of image-only retrieval performance. However, we could see that they fail to make a difference for composed image retrieval compared with our method, since the CIR task requires comprehension of multi-modalities and the interaction between visual and textual information.

---

> > ### Author Response · Authors · 2023-11-21
> > **To Reviewer K4T5**
> >
> > **Q3**: t-SNE visualizations could be shown to illustrate the differences between the different methods. This would provide a more intuitive understanding of the feature distributions and separability.
> >
> > **R3**: We present the t-SNE visualization of query and gallery features on CIRCO validation set. Specifically, we randomly sample 10 queries and their corresponding target images. The query features are extracted by the query models of different methods, while the gallery features are all extracted by the same gallery model, i.e., BLIP visual encoder. We add Figure 7 to the manuscript in the appendix. Our query features are closer to the target image features compared with Pic2word and SEARLE, which accounts for better retrieval accuracy. However, we find that for all methods, the query features and the gallery image features form two separate clusters, and the query features may be closer to other database images than their own target images in t-SNE. This could be explained by the cone effect of modality gap described in [3]: the models of different modalities create different embedding cones with very narrow range.
> > In current ZSCIR methods, the text encoder is adopted in query feature extraction and the paired visual encoder is adopted in gallery feature extraction, thus the cone effect would separate the query and gallery features. Moreover, the dimension reduction of t-SNE loses high-dimensional information, making it hard for query features to get close to their target images in t-SNE visualization.
> >
> > Generally, while the t-SNE could explain the better performance of our method, it may not be very reliable in some aspects to understand the feature distribution and separability for composed image retrieval task. This could be attributed to the cone effect of multi-modalities that makes t-SNE not as effective as in uni-modal tasks.
> >
> > [3] Liang, V. W., Zhang, Y., Kwon, Y., Yeung, S., & Zou, J. Y. (2022). Mind the gap: Understanding the modality gap in multi-modal contrastive representation learning. Advances in Neural Information Processing Systems, 35, 17612-17625.
> >
> > Finally, thank you again for your recognition and positive review to our work. We will incorporate your suggestions into our next revision.
> >
> > Note: since OpenReview does not allow image uploading in the comment box, we place the additional figures in the appendix of revised manuscript.

---

> > > ### Comment · Reviewer_K4T5 · 2023-11-22
> > > **Thank you for the response!**
> > >
> > > After reading other reviews and rebuttals, I decide to improve the final rating.

---

> > > > ### Author Response · Authors · 2023-11-22
> > > > **To Reviewer K4T5**
> > > >
> > > > Thanks for your valuable support. We will follow your suggestion and improve our final manuscript

---

### Official Review · Reviewer_WHDg · 2023-10-30

**Soundness:** 3 good
**Presentation:** 2 fair
**Contribution:** 3 good
**Rating:** 8
**Confidence:** 2

**Summary:**

The paper introduces a novel approach to composed image retrieval (CIR), emphasizing on the challenges associated with composed image retrieval that needs understanding of both visual and textual data. To address data scarcity in CIR, the authors introduce a new task paradigm named "zero-shot composed image retrieval" (ZSCIR) that transforms image retrieval to a text-to-image format, allowing for a more intuitive mapping between images and descriptive text. However, the methods presented face challenges with large-scale models which are not suitable for deployment on resource-constrained platforms, such as mobile devices. To mitigate this, the authors propose an asymmetric approach, termed Image2Sentence based Asymmetric ZSCIR, that uses different models for query and database extraction. This method utilizes a lightweight model for the user's device and a heavier model for cloud processing. The core of this approach is an adaptive token learner which converts visual features into textual tokens, thus enhancing the representation. The proposed framework was tested on various benchmarks, demonstrating its efficiency and effectiveness compared to existing state-of-the-art methods.

**Strengths:**

1. The paper addresses the challenges in Composed Image Retrieval by introducing the zero-shot composed image retrieval (ZSCIR). This method offers a fresh perspective on image retrieval by transforming it to a text-to-image format, thereby providing a more direct linkage between descriptive text and its corresponding image.

2. The introduction of an adaptive token learner, which effectively translates visual features into textual tokens, stands out as a major strength. This conversion mechanism is pivotal in enhancing the representation of images and ensuring that the retrieval process is both accurate and efficient. The adaptive nature of the learner means that it can adjust and improve over time, potentially leading to even better retrieval results in the future.

**Weaknesses:**

1. While the adaptive token learner is a strength in terms of converting visual features to textual tokens, there's a risk that the system could become overly reliant on this component. If the learner fails or encounters unanticipated scenarios, it might compromise the effectiveness of the entire retrieval process.

2. Introducing an asymmetric text-to-image retrieval approach, while innovative, adds an extra layer of complexity to the system. This might present challenges in terms of maintainability, debugging, and further development of the system.

3. The transformation of the retrieval problem from image-to-image to text-to-image inherently assumes that the descriptive texts are of high quality and detailed. Any inaccuracies or vagueness in the text could lead to inefficient or incorrect image retrievals.

4. The paper presentation is not very attractive. It is difficult to understand the novelties / contributions after reading the introduction of the paper.

**Questions:**

1.  How does the ZSCIR approach compare in performance and efficiency with state-of-the-art image retrieval methods that don't employ a text-to-image asymmetry? Are there scenarios where a traditional symmetric approach might outperform ZSCIR?

2. In terms of training data apart from the image augmentation, did you employ any data augmentation techniques to enhance the performance and robustness of the ZSCIR model? Furthermore, how did you ensure the diversity and representativeness of the descriptive texts used in the system?

---

> ### Author Response · Authors · 2023-11-21
> **To Reviewer WHDg**
>
> We appreciate the detailed comments and acknowledgment of our contributions. We provide the responses as follows.
>
> We first make a clarification of composed image retrieval task and our method. As mentioned in Introduction (section 1) in the manuscript, the task of composed image retrieval allows multi-modal queries to retrieve the images in the database, i.e., the query includes an image, and a text that describes specified modification to the image. In other words, composed image retrieval is a task of "image + text → image". In practice, the text is given by the user to represent the accurate retrieval intent. Therefore, users could adapt the text modifier to be more detailed and precise to obtain more accurate search results.
>
> For our method, we convert the "image + text → image" retrieval to the "text-to-image" retrieval by mapping the visual information to the word embedding space of the text encoder in large foundation model. During inference, the query image is converted as a pseudo sentence to concatenate with the text modifier, and they are together fed into the text encoder to extract the multi-modal query feature. The database image features are extracted by the visual encoder in large foundation model. Then by utilizing the text-image alignment capability of large foundation model, we are able to do accurate image retrieval with multi-modal query. Note that the asymmetry of our framework does not lie in the difference of modalities for the query and database side (text-to-image), but in adopting a lightweight model for the query side while adopting a large model for the database side to improve deployment flexibility and retrieval efficiency.
>
> **Q1**: While the adaptive token learner is a strength in terms of converting visual features to textual tokens, there's a risk that the system could become overly reliant on this component. If the learner fails or encounters unanticipated scenarios, it might compromise the effectiveness of the entire retrieval process.
>
> **R1**: In the context of **zero-shot** composed image retrieval, we do not have access to the labeled triplets (reference image, text modifier, target image) as direct supervision, which is necessary to train the multimodal composition capabilities of the model. Consequently, we convert visual information in the word embedding space as a descriptive pseudo sentence. This approach enables us to utilize the intrinsic composition ability of the text encoder, facilitating efficient interactions between visual information and the text modifier. This process is essential for extracting the multimodal query feature for retrieval. Therefore, it is critical to learn an effective adaptive token learner in our framework for the conversion, which is our main contribution. Our adaptive token learner is trained through global contrastive distillation loss and local alignment constraint with sufficient training data (3M), to align with the image feature space of BLIP model. Since BLIP model is pretrained on large-scale image-text dataset (129M), it possesses rich knowledge to guide the token learner to focus on prominent visual information and interact more effectively with textual information.
>
> We admit that sometimes there will be difficulty for the adaptive token learner to catch the intended target in some complicated scenarios, as shown in the failure cases of Figure 6 in the appendix. Nevertheless, in most cases, our adaptive token learner could focus on the discriminative patterns and interact well with the text modifier as shown in the good example of Figure 5 in the appendix, which plays an important role in our framework to achieve outstanding retrieval performance, as shown in Table 2,3,4 in the manuscript.
>
> In practical application scenarios, we can obtain a better token learner by enlarging the amount of training data and incorporating better multimodal foundation models to guide the learning process.
>
> **Q2**: Introducing an asymmetric text-to-image retrieval approach, while innovative, adds an extra layer of complexity to the system. This might present challenges in terms of maintainability, debugging, and further development of the system.
>
> **R2**: The lightweight visual encoders in our framework are much smaller and more efficient compared with the large foundation model, as they have fewer parameters and computations shown in Table 1. Therefore, it is much easier for maintainability, debugging and development compared with large foundation model. Although the asymmetrical structure brings about more complexity to the system, it enables faster inference for retrieval and deployment on resource-constrained platforms such as mobile phones or edge devices. Therefore, the asymmetrical structure helps to improve the retrieval efficiency. Moreover, the asymmetrical structure could also protect user privacy, since it only uploads the extracted tokens instead of real images on the Internet.

---

> ### Author Response · Authors · 2023-11-21
> **To Reviewer WHDg**
>
> **Q3**: The transformation of the retrieval problem from image-to-image to text-to-image inherently assumes that the descriptive texts are of high quality and detailed. Any inaccuracies or vagueness in the text could lead to inefficient or incorrect image retrievals.
>
> **R3**: As we mentioned earlier, different from "image-to-image" uni-modal retrieval that merely takes image as query, the task of composed image retrieval allows multi-modal queries to retrieve the images in the database, which is a task of "image + text → image". In practice, the text is given by the user to represent the accurate retrieval intent. Therefore, users could adapt the text modifier to be more detailed and precise to obtain more accurate search results.
>
> In our framework, we convert the "image + text → image" problem to the "text-to-image" problem by mapping the visual information to the word embedding space of the text encoder in large foundation model as pseudo sentence. Therefore, the visual information could interact with the text modifier more efficiently in the text domain to obtain the multimodal composed features. We further utilize the text-image alignment capability of large foundation models to do accurate image retrieval, and achieve the best performance on three benchmarks.
>
> **Q4**: The paper presentation is not very attractive. It is difficult to understand the novelties / contributions after reading the introduction of the paper.
>
> **R4**: We are sorry for that we did not present clear contributions in the manuscript. Here we summarize our contributions as follows:
>
> 1. We propose a new framework for **zero-shot** composed image retrieval, which solves composition learning without the expensive labeled-triplets for supervised training.
>
> 2. We propose an asymmetrical structure for composed image retrieval to enable flexible deployment on resource-constrained platforms and improve the retrieval efficiency.
>
> 3. We propose an adaptive token learner to map image as a sentence in the word embedding space to interact with text information, and automatically filter out noisy visual information to preserve discriminative semantics.
>
> 4. Global constrastive distillation and local alignment regularization are proposed to guide the lightweight encoder learning with the knowledge of large foundation model, and our method achieves significant retrieval performance on CIRR, CIRCO and FashionIQ datasets.
>
> **Q5**: How does the ZSCIR approach compare in performance and efficiency with state-of-the-art image retrieval methods that don't employ a text-to-image asymmetry? Are there scenarios where a traditional symmetric approach might outperform ZSCIR?
>
> **R5**: As report in Table 2,3,4 in the manuscript, our asymmetrical method outperforms the state-of-the-art methods Pic2word and SEARLE in terms of retrieval performance, which adopt symmetrical structure. In Table 1 in the manuscript, we report the model parameter size and computational consumption for different query models. Here we consider Pic2word and SEARLE adopting BLIP VE as both query and gallery model. Compared with the symmetrical structure that adopts large visual encoder (BLIP VE), lightweight models require much fewer computational resources. Furthermore, we report the latency on both query side and cloud side on CIRCO validation set. For query side, we report the inference latency for different query models in the following table. We find that lightweight encoders are generally twice faster than the large visual encoder (BLIP VE). For cloud side, since the gallery models are the same (BLIP TE & VE), the retrieval latency is the same. Generally, the lightweight encoders are much more efficient than the heavy foundation model in terms of resource consumption and inference latency.
>
> | Query model | Query inference latency (ms, per query) | Gallery model | Retrieval latency (ms, per query) | Total retrieval latency (ms, per query) |
> | :------:| :------:| :------: | :------: | :------: |
> | EfficientNet B0 | 3.07 | BLIP VE | 3.58 | 6.65 |
> | EfficientNet B2 | 3.24 | BLIP VE | 3.58 | 6.82 |
> | EfficientViT M2 | 3.27 | BLIP VE | 3.58 | 6.85 |
> | MobileNet V2 | 3.03 | BLIP VE | 3.58 | 6.61 |
> | MobileViT V2 | 3.40 | BLIP VE | 3.58 | 6.98 |
> | BLIP VE | 6.53 | BLIP VE | 3.58 | 10.11 |
>
> To train the symmetrical framework, the large visual encoder needs to be updated during training. Since the large visual encoder is much heavier than our lightweight encoder, it is more likely to overfit and requires larger GPU memory to reach the same batch size for good performance during training. With more training data and more GPU resources, we believe that the symmetrical setting would outperform the asymmetrical structure in terms of retrieval accuracy.

---

> > ### Author Response · Authors · 2023-11-22
> > **To Reviewer WHDg**
> >
> > **Q6**: In terms of training data apart from the image augmentation, did you employ any data augmentation techniques to enhance the performance and robustness of the ZSCIR model? Furthermore, how did you ensure the diversity and representativeness of the descriptive texts used in the system?
> >
> > **R6**:  In our framework, to avoid reliance on expensive triplet training data, we only use image data for training to adhere to the requirement of zero-shot scenario. Therefore, only image augmentation is used during training, including radom flip and crop.
> >
> > As mentioned in R3, the text descriptions are given by the users and do not require any specific format, which describe the heterogenous users' intents for retrieval and inherently possess sufficient diversity. Users could adjust the text information to be more precise and accurate to obtain better retrieval results.
> >
> >
> > Finally, thank you again for your thoughtful comments. We will incorporate your suggestions into our next revision. If some of your concerns are addressed, you could consider raising the rating. This is very important for us and we will appreciate it very much.
> >
> > Note: since OpenReview does not allow image uploading in the comment box, we place the additional figures in the appendix of revised manuscript.

---

> ### Comment · Reviewer_WHDg · 2023-11-22
> **Thanks for the detailed rebuttal**
>
> I thank the authors for writing very detailed rebuttal and reply all the weaknesses and questions that I mentioned. After looking into the rebuttal and also going through other reviews and replies, I think the paper is making a good contributions to the community. So I have decided to increase my rating. I hope the authors would address the comments in the final version of the paper.

---

> > ### Author Response · Authors · 2023-11-22
> > **To Reviewer WHDg**
> >
> > Thanks for your valuable support and raising the rating. We will follow your suggestion and incorporate the results of the discussion to refine our final manuscript

---

### Official Review · Reviewer_ZKF9 · 2023-11-06

**Soundness:** 3 good
**Presentation:** 3 good
**Contribution:** 3 good
**Rating:** 6
**Confidence:** 4

**Summary:**

This paper proposes an image2sentence based asymmetric framework for zero-shot composed image retrieval tasks. In particular, a lightweight visual encoder and a consequent adaptive token learner are designed to effectively extract the visual features from query images for the mobile side. By doing so, the learned features could be generated as a good visual prompt as with the text intent for conventional LLM to deal with image retrieval tasks. In addition, a local alignment regularization term is added to further improve the training. The experiments conducted on several benchmark datasets verify the effectiveness of the proposed method compared with existing SOTA ones.

**Strengths:**

1. This paper is of good written quality that makes the readers easy to follow. The logic, the notion expression, and the experiments are all very clear.

2. The asymmetric design is interesting and this design has been proven an efficient way to deal with resource-limited circumstances.

3. The experiments conducted are very convincing to support the contribution claimed by the authors. The properties of the proposed method are well demonstrated in the ablation study.

**Weaknesses:**

1. It could be better to discuss more about the number of token selections in detail. According to Fig 3, it seems the performance is a bit sensitive to the number of tokens used in the proposed method. Then, a more detailed discussion of this observation with a visual example of the same query but a different number of tokens could help to better demonstrate the impact brought by the tokens.

2. It could be good to add a discussion of the relationship between the proposed adaptive token learner and the similar approach used in the following papers [1,2]. They have a similar structure, a discussion would help to better locate the position of the token learned in this work.

 [1] Wu, Hui, et al. "Learning token-based representation for image retrieval." Proceedings of the AAAI Conference on Artificial Intelligence. Vol. 36. No. 3. 2022.
 [2] Locatello, Francesco, et al. "Object-centric learning with slot attention." Advances in Neural Information Processing Systems 33 (2020): 11525-11538.

**Questions:**

Please check the weaknesses above.

**Details Of Ethics Concerns:**

Nil

---

> ### Author Response · Authors · 2023-11-21
> **To Reviewer ZKF9**
>
> We appreciate the detailed comments and acknowledgment of our contributions. We provide the responses as follows.
>
> **Q1**: It could be better to discuss more about the number of token selections in detail. According to Figure 3, it seems the performance is a bit sensitive to the number of tokens used in the proposed method. Then, a more detailed discussion of this observation with a visual example of the same query but a different number of tokens could help to better demonstrate the impact brought by the tokens.
>
> **R1**: Generally, to achieve accurate retrieval, the pseudo sentence tokens need to accomplish two aspects: extracting effective visual information and interacting with textual information. When the token length is very small, there would be drop in performance due to the lack of capacity to extract sufficient visual information. As shown in Figure 8 in the appendix of manuscript, query model with very small token length performs worse even if the attention maps are reasonable.
>
> However, when the token length increases for too long, on one hand, some tokens may focus on trivial and noisy visual patterns; on the other hand, these tokens may interact incorrectly with the text modifier, and may impact the correct token-text interaction. In Figure 4 in the manuscript, we have demonstrated the comparison of different token lengths with the same query. With the visualization of the attention maps of adaptive token learner and the retrieval results, it could be seen that some sentence tokens may tend to focus on the background or the trivial patterns with excessively large token lengths, which would likely introduce noise to degrade the retrieval performance. We further show more comparisons of different token lengths, including the retrieval results, attention maps of sentence tokens with text modifier and the visual feature map in Figure 9 in the manuscript appendix. As shown in Figure 9, the excessive tokens associate the visual patterns with incorrect text information, potentially interfering with the interaction of multi-modal information. Even if there are accurate associations between other sentence tokens and text modifier, this interference would still degrade the retrieval accuracy.
>
> **Q2**: It could be good to add a discussion of the relationship between the proposed adaptive token learner and the similar approach used in the following papers [1,2]. They have a similar structure, a discussion would help to better locate the position of the token learned in this work.
>
> **R2**: In [1], the token learner is adopted to jointly learn local feature representation and aggregation in a unified framework. The output of token learner is the local features, which are concatenated as the aggregated global feature for image retrieval. In [2], the token learner maps a set of N input feature vectors to a set of K output vectors as slots, each of which may describe an object or entity in the input for the task of object discovery and set prediction.
>
> As pointed out in [1] and [2], the token learner is utilized to focus on multiple discriminative visual patterns or entities and filter noise information such as background and indiscriminative image regions. Therefore, this structure could serve as an efficient visual extractor in various visual tasks.
>
> Although our method also uses adaptive token learner, the semantics of token learner output are different from theirs. The output of our token learner resides in the word embedding space of the text encoder, which serves as a pseudo sentence, with each token acting like a word to interact with information of text modifier for further multimodal composition in CIR task. While in [1] and [2], the outputs are used directly for uni-modal vision tasks like similarity retrieval and object recognition. Therefore, the utilization of token learner in [1] and [2] is not suitable for our composed image retrieval task, while our token learner is devised for this task and achieves promising retrieval results in experiments.
>
> [1] Wu H, Wang M, Zhou W, et al. Learning token-based representation for image retrieval. Proceedings of the AAAI Conference on Artificial Intelligence. 2022, 36(3): 2703-2711.
>
> [2] Locatello, Francesco, et al. "Object-centric learning with slot attention." Advances in Neural Information Processing Systems 33 (2020): 11525-11538.
>
>
> Finally, thank you again for your recognition and positive review to our work. We will incorporate your suggestions into our next revision.
>
> Note: since OpenReview does not allow image uploading in the comment box, we place the additional figures in the appendix of revised manuscript.

---

### Meta-Review · Program_Chairs · 2023-12-05

**Metareview:**

This work addresses composed image retrieval (CIR), focusing on mobile applications. The authors propose to convert CIR to a text-based image retrieval problem by extracting a descriptive sentence from the input image. They also use a lightweight model for the query to be more amenable to mobile applications. All reviewers found the work interesting, and the experimental validation convincing. The authors clarified some misunderstandings and improved the paper presentation after discussions with the reviewers. The reviewers now unanimously recommend acceptance. The AC sees no reason to override this recommendation.

**Justification For Why Not Higher Score:**

Narrow scope for ICLR audience.

**Justification For Why Not Lower Score:**

See above

---

### Decision · Program_Chairs · 2024-01-16

Accept (spotlight)